# HARPA: A Testability-Driven, Literature-Grounded Framework for Research Ideation

## Abstract

While there has been a surge of interest in automated scientific discovery (ASD), especially with the emergence of LLMs, it remains challenging for tools to generate hypotheses that are both testable and grounded in the scientific literature. Additionally, existing ideation tools are not adaptive to prior experimental outcomes. We developed HARPA[1] to address these challenges by incorporating the ideation workflow inspired by human researchers. HARPA first identifies emerging research trends through literature mining, then explores hypothesis design spaces, and finally converges on precise, testable hypotheses by pinpointing research gaps and justifying design choices. Our evaluations show that HARPA-generated hypothesis-driven research proposals perform comparably to a strong baseline AI-researcher across most qualitative dimensions (e.g., specificity, novelty, overall quality), but achieve significant gains in feasibility(+0.78, p< 0.05, bootstrap) and groundedness (+0.85, p< 0.01, bootstrap) on a 10-point Likert scale. When tested with the ASD agent (CodeScientist), HARPA produced more successful executions (20 vs. 11 out of 40) and fewer failures (16 vs. 21 out of 40), showing that expert feasibility judgments track with actual execution success. Furthermore, to simulate how researchers continuously refine their understanding of what hypotheses are both testable and potentially interesting from experience, HARPA learns a reward model that scores new hypotheses based on prior experimental outcomes, achieving approx. a 28% absolute gain over HARPA's untrained baseline scorer. Together, these methods represent a step forward in the field of AI-driven scientific discovery.

## 1 Introduction

Scientific discovery fundamentally depends on effective hypothesis generation—a creative, iterative, and cognitively complex process. In the past year, advances in large language models (LLMs) have revitalized the field of Automated Scientific Discovery (ASD) and AI-assisted ideation, by providing the foundations for agents that can autonomously execute experiments (Lu et al., 2024; Gottweis et al., 2025; Jansen et al., 2025; Li et al., 2024c). At the same time, these models have been applied to generate novel research ideas (Radensky et al., 2024; Pu et al., 2024; Baek et al., 2024b; Wang et al., 2023; Li et al., 2024b), supplying candidate ideas for the experimental agents to explore.

One of the central challenges of automated scientific discovery is that the hypotheses generated by large language models rarely rise to the level of breakthrough discoveries (Gottweis et al., 2025). While such hypotheses may be novel or creative, they are frequently infeasible as research proposals (Si et al., 2025). Common issues include limited grounding in literature, omission of critical methodological details, and reliance on resource-intensive experimental designs that exceed the capacity of ASD agents. These challenges mirror findings from prior studies, where ideation systems often produce ideas that are too abstract to be actionable, require substantial human intervention to refine into testable research proposals (Li et al., 2024b; Vasu et al., 2025; Radensky et al., 2024; Wang et al., 2023; Pu et al., 2024), or lack mechanisms to balance novelty with feasibility (Li et al., 2024c; Jansen et al., 2025; Gottweis et al., 2025).

In this work, we present HARPA — **H**ypothesis & **R**esearch **P**roposal **A**ssistant — a novel multi-stage computational framework that generates literature-grounded research proposals with specific hypotheses well-supported for ASD systems. HARPA is composed of a *proposal generator* and a *scorer*, as shown

---

[1]All code and data used in this paper will be made publicly available at GitHub Link: (removed for review).

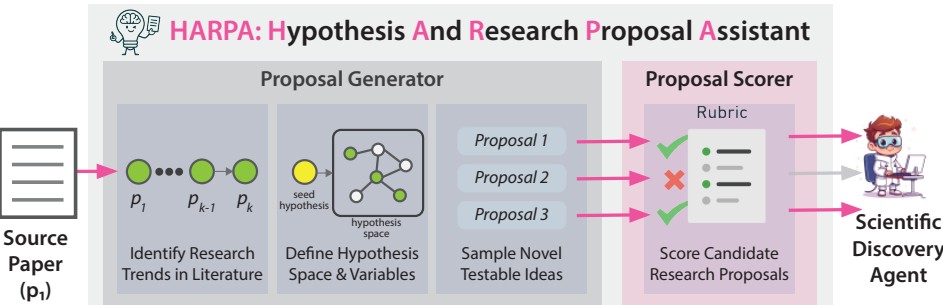

Figure 1: Overview of HARPA. Starting from a source paper's abstract, seed hypothesis derived from literature trends, HARPA constructs a *world model* of variables, values, and supporting evidence. The proposal generator consists of three stages (trend identification, hypothesis space exploration for divergence, proposal sampling for convergence) to produce candidate hypothesis-driven research proposals. A dedicated scorer employs reasoning-based reward model based on prior execution evidences to evaluate testability w.r.t target ASD agent.

in Figure 1. HARPA generates significantly more feasible research ideas by virtue of its generation approach being strongly grounded in the scientific literature: first identifying relevant research trends related to a user's hypothesis of interest, then systematically exploring the constructed hypothesis space of variables and their values, and finally converging on novel (and significantly more feasible) hypotheses as well-reasoned combinations of variables and research trends that fill identifiable research gaps in the literature.

We evaluate HARPA using a combination of expert human studies as well as ASD agents. We empirically show that HARPA-generated proposals are more feasible and better grounded in the scientific literature than those produced by contemporary systems. Beyond these gains, we further demonstrate that HARPA's reward-trained scorer, distilled in RM-R1 style (Chen et al., 2025c), can predict which research proposal is most likely to execute on the ASD agent. Unlike a black-box classifier, the scorer produces rubric-style reasoning traces, interpretable justifications generated from the proposal content and conditioned on the ASD agent's capabilities, trained to reflect patterns distilled from prior execution outcomes. This enables HARPA to incorporate feedback from prior experimental evidences to selectively generate proposals tailored to the strength and constraints of specific ASD agent - much as a professor might guide a student toward research ideas aligned with the student's prior knowledge and expertise.

Our results empirically demonstrate that HARPA nearly doubles the scientific output of automated discovery systems, measured as the number of successfully executed experiments, while also reducing costs by pruning infeasible proposals unlikely to succeed before they are attempted. Our contributions:

- **HARPA:** a novel literature-grounded framework for hypothesis generation for ASD systems, that combines identifying research trends, hypothesis space construction, and testability-aware convergence to generate proposals that are novel and executable.
- **Empirical demonstration:** studies with both human domain experts and automated scientific discovery systems showing that HARPA-generated proposals are rated higher in feasibility (+0.78, $p \leq 0.05$) and literature-grounding (+0.85, $p \leq 0.01$), and achieve higher execution success ($\sim$ 80% more, 20 vs. 11), compared to competing systems.
- **Learned feasibility:** We demonstrate that HARPA's scorer — an interpretable reward model distilled from actual execution traces, can predict which research proposals are most likely to be executable by a given ASD agent, significantly saving time and cost by selectively pruning hypotheses that the system is unlikely to execute. HARPA's scorer achieves a +0.28 absolute, 53% relative gain over the untrained baseline scorer.
- A publicly available implementation of this approach (HARPA), including the HARPA-Scorer model (to be released on Hugging Face), and first large-scale ASD execution traces and preference dataset to support reproducibility and future research.

Together, these contributions represent a step toward more capable hypothesis generation tools and help advance the rapidly growing field of AI-driven scientific discovery.

## 2 RELATED WORK

**Human hypothesis generation.** Cognitive science highlights that scientific hypothesis generation is a complex iterative process involving strategies such as analogical reasoning and model based thinking, where simplified representations guide inquiry (Dunbar, 2000; Nersessian, 2010; Klahr & Simon,

1999). The Scientific Discovery as Dual Search (SDDS) model (Klahr & Dunbar, 1988) identifies strategies such as searching memory for relevant hypotheses and generalizing from experimental results, underscores the need for the ASD systems that can reason over structured hypothesis spaces and adapt from experimental feedback. Prior work has also examined how researchers navigate the broader scientific landscape, where scientists often favor incremental, topic-adjacent experiments (Rzhetsky et al., 2015), with only a minority pursuing riskier but higher-impact directions (Foster et al., 2015).

**Automated ideation tools.** Computational frameworks such as Literature-Based Discovery (LBD) (Swanson, 1986) illustrate how disconnected literatures can be bridged to reveal hidden hypotheses. More recent systems (Radensky et al., 2024; Wang et al., 2023) focus on producing super-brief, novelty-driven research ideas typically assessed with human judgments rather than execution. Systems such as Chain of Ideas (Li et al., 2024b) and HypER (Vasu

| System | 1) Grounded ideas? | 2) Domain-General? | 3) Full proposal? | 4) ASD Feasibility? | 5) Adaptive? |
|---|---|---|---|---|---|
| GPT-5 | × | ✓ | ∼ | × | × |
| Scideator | ✓ | ✓ | × | × | × |
| Moose-Chem | ✓ | × | × | × | × |
| CodeScientist | ✓ | ✓ | ✓ | × | × |
| AI researcher | ✓ | ✓ | ✓ | × | × |
| **HARPA (ours)** | ✓ | ✓ | ✓ | ✓ | ✓ |

Table 1: Comparison of ideation systems in terms of: 1) Are the ideas grounded in related work? 2) Can the ideator generate open-domain ideas? 3) Generates brief ideas or full proposal? 4) Does it consider feasibility w.r.t ASD agents? 5) Does it learn from prior experiments? (✓: yes, ×: no, ∼: sometimes).[2]

et al., 2025) identify literature trends but generate ideas that are too high-level to be actionable, while Scideator (Radensky et al., 2024) generates diverse coarse-grained facets such as purpose, mechanism, or contribution, offering novelty but lacking operational clarity and require human refinement. IdeaSynth (Pu et al., 2024) transforms research ideas into proposals but demands substantial human-in-the-loop involvement, limiting scalability. Recent systems such as Nova (Hu et al., 2024), Dolphin (Yuan et al., 2025), and hypothesis proposers (Yang et al., 2024a; Qi et al., 2023) further explore enhancing novelty and diversity in LLM-generated ideas, but they remain too high-level to be actionable for proposal-level evaluation. Existing systems lack mechanisms to adapt their ideation in response to experimental feedback (Table 1). In this paper, we compare HARPA with the AI Researcher method (Si et al., 2024), which was custom-built for open-domain proposal generation and has demonstrated state-of-the-art performance on this task.

**Bridging ideation and execution.** Large-scale evaluations (Si et al., 2024; 2025) show that while AI-generated ideas may be perceived as more novel than expert-authored ones, they are often less feasible experimentally. Other ideation frameworks, including MLR-Copilot (Li et al., 2024c) and Agent Laboratory (Schmidgall et al., 2025), emphasize benchmark-guided or multi-agent settings but fall short of systematic experimental comparisons. Execution focused systems like CODE SCIENTIST (Jansen et al., 2025) and AI-Scientist (Lu et al., 2024) demonstrate end-to-end automated experimentation but assume hypotheses are already well-structured and feasible. NovelSeek (Team et al., 2025) extends this direction by performing multi-round optimization and debugging given an existing idea or codebase, rather than generating open-domain full proposals. HARPA complements these systems by generating structured, literature-grounded proposals. Unlike other ideators, HARPA integrates a reward model conditioned on ASD capabilities, making research hypotheses generation novel, grounded, and experimentally feasible. This makes it useful for human researchers positioning it as a building block toward the long-term vision of "robot scientists" (King et al., 2009). Table 1 compares HARPA with representative systems in the literature over different ideation attributes.

## 3 HARPA: HYPOTHESIS AND RESEARCH PROPOSAL ASSISTANT

HARPA's design is inspired by studies of how humans generate hypotheses (Section 2). HARPA consists of two core components: a *proposal generator* and a *scorer*. The proposal generator begins with a user-given source paper and generates detailed, literature-grounded hypothesis-driven research proposals by treating hypotheses as structured research artifacts, enriched with a rationale (literature-based justification explaining how prior work motivates the preliminary hypothesis), related work, key variables, and operationalization plans. The scorer complements this process by ranking and filtering proposals with a learned reward model that predicts feasibility and testability without requiring full execution. These components together allow HARPA to produce hypothesis-driven proposals that are not only novel and grounded in prior work, but also prioritized for practical execution by ASD agents.

---

[2]Systems: GPT-5 (OpenAI, 2025), Scideator (Radensky et al., 2024), Moose-Chem (Yang et al., 2024b), CodeScientist Ideator (Jansen et al., 2025), AI-Researcher (Si et al., 2024)

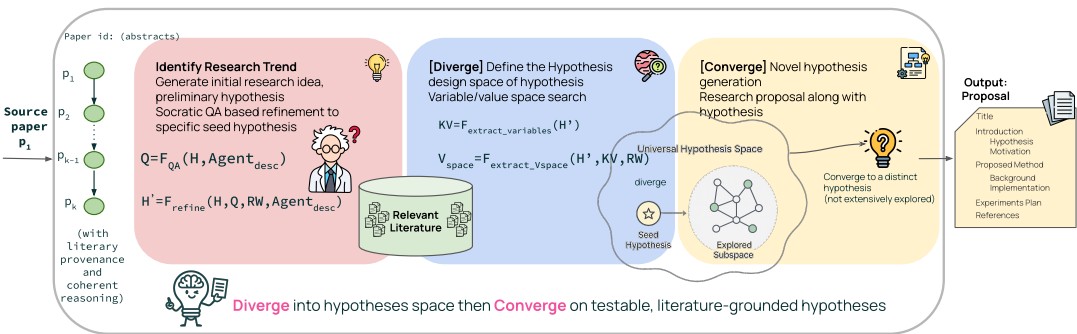

Figure 2: HARPA's Proposal Generator: Divergence and convergence to literature grounded novel proposals

## 3.1 HARPA's Proposal Generator

HARPA begins by constructing the scientific reasoning chain of papers given a source paper. The chain construction is based on (Vasu et al., 2025), where each paper is connected to the previous paper based on its scientific dependency and the citing relation. This reasoning chain enables HARPA to identify a preliminary research gap and the motivation to come up with a research problem and hypothesis (Appendix L 8). However, the seed hypothesis generated at this stage is not very specific and is not optimized for novelty or the feasibility of the idea. To systematically develop a literature-grounded research proposal that is also novel and feasible, we follow the following steps:

**Refinement with Socratic Question Answering** Recently, Socratic questioning has been applied to language models as a self-guiding mechanism (Chen et al., 2025b). We apply Socratic questioning to help the model think critically, uncover assumptions, and prompt a deeper understanding of the generic preliminary hypothesis. Given a set of relevant literature snippets ($RW$) extracted from related works associated with this preliminary hypothesis, and the description of the underlying ASD agent ($Agent_{desc}$) which executes this hypothesis ($H$), we generate a set of at least 20 questions, $\mathcal{Q}$, that helps to navigate the specificity of this hypothesis (see Appendix L 10). This is denoted as $\mathcal{Q} = \mathcal{F}_{QA}(H, Agent_{desc})$. Then, given this set of $\mathcal{Q}$ and the set of literature snippets $RW$ relevant to $H$ and $Agent_{desc}$, the language model can refine $H$ by answering these $\mathcal{Q}$. We denote this using $H' = \mathcal{F}_{refine}(H, \mathcal{Q}, RW, Agent_{desc})$. The detailed instruction to refine and make the hypothesis more specific is given in Appendix L 11. Using $H$ as the query, we systematically collect $RW$ using the snippet search over S2ORC corpus (Singh et al., 2025). Specifically, $H$ is progressively generalized $n$ times (see Appendix L 9) and each version of $H$ is used as a query to collect $RW$.

**Defining the hypothesis design space using $H'$** To understand the key concepts and variables around this hypothesis in hand, we first extract the set of key variables ($KV$) from it (see L12), denoted as $KV = \mathcal{F}_{extract\_var}(H')$. However, the relevant literature might have already explored similar variables or different values of these variables (a brief illustration in Appendix L 7). We extract and define this key variable space as $\mathcal{V}_{space} = \mathcal{F}_{extract\_space}(H', KV, RW)$. In this process, we ensure that each of these key variables or values mentioned in the related literature is associated with $H'$. To do this, the extraction process also extracts metadata such as the source paper title and the specific details and description relevant to $H'$ about this variable (see Appendix L13). We also allow the model to add as extra variables a small set of standard evaluation metrics ("accuracy", "precision") that were given as illustrative examples in the prompt. When these are added without direct literature evidence, they are explicitly marked as "LLM-recommended".

**Convergence to novel hypothesis** The research trend, initial idea, and hypothesis design space can be seen as HARPA's world model around the initial idea $H'$. It encodes the key components of the idea and if or how they are being addressed in the relevant literature. Given this hypothesis space, $H'$ is now converged into a distinct hypothesis, $H_{final}$, which has not been extensively studied in the given space. We denote this process as $H_{final} = \mathcal{F}_{generate}(H', \mathcal{V}_{space}, RW)$. Along with $H_{final}$, we also generate the detailed description of this hypothesis such as overview, detailed description of key variables, idea design including how the combination of the variables can be integrated or how the hypothesis can be implemented in a high level and some of the source papers (initial chain of papers and trend) from which

Figure 3: **HARPA Scorer:** *1. Training Data Generation.* HARPA generates candidate proposals $(P_a, P_b)$, which are executed in the ASD-agent environment to produce raw execution traces $(E_a, E_b)$. A teacher LLM analyzes these traces and outputs a high-fidelity rubric-style reasoning trace with justification and answer $(Reason\_trace(P_a, P_b))$. *2. Reasoning Distillation and Reward Modeling.* The student model is distilled from these reasoning traces, initialized as a policy, and fine-tuned via RLVR using preference labels to produce a rubric-style reasoning trace and a preference label (e.g., "Proposal A wins", an example trace in Appendix L 1).

this idea is evolved as related work (detailed instruction in L14, Appendix F). All LLM function calls in this pipeline were backed by `GPT-4o model`.

HARPA also specifies the operationalization of this idea, so that the underlying ASD agent or human researcher can have more details about its implementation plan. For this, we utilize the functionality—`idea to implementation plan`—of CodeScientist (Jansen et al., 2025). All this information together forms the final $\mathcal{HARPA}_{proposal}$ (example in Appendix C).

### 3.2 HARPA SCORER: ESTIMATING TESTABILITY OF PROPOSALS

Generating and executing every candidate proposal, whether by human researchers or autonomous agents, is infeasible at scale. To address this challenge, we develop a *learned reward model* that predicts the likely success of a research proposal without requiring full execution. Existing approaches either rely on direct execution (costly) (Li et al., 2024c; Lu et al., 2024) or on heuristic judgments by LLMs on feasibility (Si et al., 2024; Chen et al., 2025a; Yang et al., 2024b; Baek et al., 2024a), which are often unreliable (Li et al., 2024a) and lack grounding in prior experimental evidence (Zhu et al., 2025). Our goal is to provide a scalable and interpretable mechanism to filter and rank research proposals, prioritizing those that are both novel and feasible for the given ASD agent. See Figure 3 for the overview of the HARPA scorer.

**Training Data Generation.** We collect preference data by executing HARPA-generated proposals using an off-the-shelf ASD agent, CODESCIENTIST, that runs containerized Python experiments. Each execution ($E$) produces raw traces of the experiment setup, intermediate errors, and automatic assessments, and a final report. We convert the structured experiment summaries (e.g., Appendix L 2) generated by CODESCIENTIST into categorical outcome labels using a meta-analysis scheme:

$$\text{Label}(E) = \begin{cases} \text{Success} & \text{if faithfulness\_category} = \text{faithful}, \\ \text{Failure} & \text{if faithfulness\_category} = \text{errors} \\ & \quad \vee \text{ (faithfulness\_category} = \text{inconclusive} \wedge \\ & \quad \text{hypothesis\_category} = \text{inconclusive}), \\ \text{Uncertain} & \text{otherwise.} \end{cases}$$

where faithfulness\_category indicates whether the experiment was executed faithfully without implementation errors, and hypothesis\_category captures whether the observed outcomes 'support,' 'reject,' or remain 'inconclusive' w.r.t the original hypothesis. The *Uncertain* label captures executions that neither cleanly succeed nor fail, ensuring ambiguous traces do not distort the success or failure boundary. These labels are then used to construct pairwise preferences: for each pair $(P_a, P_b)$, a teacher LLM analyzed (see Appendix L 3) the corresponding traces and generated a rubric-style reasoning trace, along with a preference judgment based on the observed outcome. This yields high-quality training data for the distillation, consisting of pairwise comparisons with interpretable justifications that reflect the empirical feasibility.

**Reasoning Distillation and Reward Modeling.** We train the HARPA scorer in two stages following the RM-R1 framework (Chen et al., 2025c). First, we distill the teacher's rubric-style reasoning traces into the student model. This facilitates the student with the ability to generate interpretable justifications

aligned with teacher rubrics. Next, we train the distilled model with preference-based optimization using the RLVR strategy (Chen et al., 2025c), aligning its scoring with empirically verifiable outcomes ('success,' 'failure') from CODESCIENTIST executions. The model outputs both (i) a comparative label (e.g., "Proposal A wins" ) and (ii) a rubric-style reasoning trace explaining the decision. This dual output allows the model to function not only as a black-box scorer but also as an explainer, providing transparent, human-readable justifications that can be used to refine research proposals. An example reasoning trace is provided in Appendix L 1, showing how the model assigns higher feasibility to one proposal using execution-derived factors, such as execution success, complexity (based on reflection), and cost efficiency parsed from the structured experiment summary of CODESCIENTIST.

**Conditioning on ASD capabilities.** To ensure judgements are adaptable to the targeted execution environment, the reward model is conditioned on an explicit ASD agent (see Appendix L 3), specifying constraints such as compute budget, permissible evaluation protocol, dataset access, and whether human involvement is allowed. During both training and inference, the agent profile is concatenated with the proposals and execution metadata. In our case, conditioning reflects the limits of CODESCIENTIST, but the same mechanism applies to other agents. For instance, proposals requiring human studies or private datasets are down-ranked for CODESCIENTIST but could go higher for a more capable agent. This makes HARPA's scorer adaptive, producing feasibility-aware rankings that generalize across different discovery settings.

## 4 EXPERIMENTS

We evaluate HARPA along two complementary axes: (1) a human-centric expert study to evaluate whether generated proposals are appealing to human researchers, and (2) an ASD-centric execution study, which measures the operational testability of proposals through the reward modeling.

### 4.1 BASELINES

We compare HARPA against different baselines depending on the evaluation axis. **Human-centric Evaluation:** We compare HARPA proposal generator against AI-Researcher (Si et al., 2024), a strong baseline for literature-grounded ideation. We standardized section headings to match proposal formats across systems. For references, we included the papers AI-Researcher internally retrieved, whereas HARPA had literature identified during its multi-stage pipeline. To ensure comparability, we generated topics from each source paper's abstract (since AI-Researcher expects a topic rather than a source paper). Both systems are seeded with the same information (abstract of the source paper), and neither is given access to the full paper. Each system then uses its own retrieval pipeline, which is an integral part of its design and therefore kept unchanged for end-to-end comparison. Apart from this topic generation step, all other settings followed the original AI-Researcher implementation. **Agent-centric Evaluation:** For the HARPA scorer, we compare the two variants: (i) an untrained LLM scorer applied directly to a pair of proposals, and (ii) the HARPA scorer, our distilled and RLVR-trained reward model. This setup allows us to isolate the benefit of training the scorer while keeping the proposal generator fixed. We use `Qwen-7B-Instruct` as the backbone, with the non-finetuned model as the LLM scorer baseline and the trained version as HARPA scorer.

### 4.2 HUMAN-CENTRIC EVALUATION SETUP

**Participants:** We recruited 12 experts who have experience in writing and reviewing scientific articles in their domain of interest via *Upwork.com*. See Appendix A.1 for detailed backgrounds and screening criteria. **Dataset:** Our evaluation corpus was constructed dynamically by the experts themselves. Each expert selected source papers ($\geq 20$ citations, published before 2025) in their domain of expertise. This design ensured informed and fair evaluation in a familiar context. For each source paper, we generated two proposals from HARPA and two from the baseline, and each expert evaluated proposals from at most two source papers of their choice. This process resulted in 40 proposals per system overall. Proposals were uniformly formatted with identical section headings — `title`, `introduction`, `proposed method`, `experiments plan`, and `references`, and covered diverse topics (e.g., NLP, RAG, RL, Optimization). (Corpus statistics in Appendix A.1.1)

**Evaluation Rubric:** We adapted our evaluation rubric from the idea review form of Si et al. (2024) for evaluating research proposals. Experts rated each proposal on a 10-point Likert scale for *Familiarity, Novelty, Feasibility, Expected effectiveness, Excitement, Overall, and Confidence*, providing brief

textual justifications (full rubric in Appendix B). In addition to the original rubric, we introduced four dimensions relevant to hypothesis-driven proposals and their operationalization: *Literature Grounding, Motivation from Literature, Coherence of Idea Composition, and Specificity of Proposed Method*. In total, the rubric covered 11 dimensions, with full wording provided in Appendix B. **Protocol:** Proposals were presented to each expert in randomized order, with system identities hidden. The same expert who provided the source paper independently assessed and rated all four proposals (including baseline and HARPA) to ensure fair comparison on the same topic. Some experts reviewed proposals for more than one source paper [3]. Data collection was carried out using the Label Studio platform and experts were compensated at a rate of 35USD/hr.

### 4.3 AGENT-CENTRIC EVALUATION SETUP

Here we evaluate proposals by executing them with the CODESCIENTIST providing data.

**Data Curation.** We sampled 275 highly cited ACL papers as source papers and generated up to five HARPA proposals per paper ($1,222$ total). Each proposal was executed *five* times each in CODESCIENTIST to avoid the stochasticity in LLM-based code generation. From each of the five runs, we selected the execution trace that most truly representing the research proposal and considered that for further analysis. Outcomes were labeled as SUCCESS ($29.38\%$), FAILURE ($51.55\%$), or UNCERTAIN ($19.07\%$) according to the categorical outcome labels described before.

**Preference Construction and Training:** From these labeled executions, we constructed 3954 preference pairs on shared source paper topic (see Appendix L 5). Each pair with execution metadata was used to generate a rubric-style reasoning trace (including preference judgments) by an oracle model[4], which achieved $87.48\%$ accuracy. We filtered the pairs with correct judgements (3459) and their reasoning traces as ground truth for further experiments. **Distillation and RLVR:** We split the proposals into training (2595), validation (452), and test (412) subsets. Following the RM-R1 framework Chen et al. (2025c), we first distilled a student model to generate interpretable rubric-style reasoning aligned with teacher rubrics. We further applied RLVR training on preference pairs (using an additional 226,170 success-failure pairs irrespective of shared topic). Finally, we evaluated the distilled reward model on a held-out set of success-failure pairs (186), using accuracy and qualitative analysis of reasoning traces. This two-stage process yields the *HARPA Scorer* that is both interpretable and adaptive to ASD execution (more implementation details in Appendix D.1).

A full end-to-end evaluation would be ideal, but it is too expensive and would require generating many additional proposals and obtaining impractical expert annotation on largely random samples. We evaluate the two components separately, since the scorer only becomes meaningful once the generator produces executable hypotheses: expert review on a small set of relevant samples and large-scale testability on diverse ACL papers. This setup provides a controlled evaluation of each component in isolation, and we leave a full end-to-end evaluation to future work.

## 5 MAIN RESULTS

### 5.1 HUMAN-CENTRIC RESULTS

Figure 4 summarizes the expert evaluations of HARPA's proposal generator against the baseline across 11 dimensions. Nine dimensions define the research proposal quality (i.e., novelty, feasibility, expected effectiveness, excitement, grounding, specificity, coherence, motivation, and overall quality), while two meta-dimensions capture the user's familiarity with the proposal topic and their confidence in the judgment (complete proposal evaluation form in Appendix B). HARPA shows statistically significant gains in feasibility ($+0.78$, $p < 0.05$, bootstrap) and grounding ($+0.85$, $p < 0.01$, bootstrap). For specificity, motivation, and overall scores, HARPA shows a positive trend, although it does not rise to the level of statistical significance. For other metrics, HARPA performs comparably to the baseline (Appendix Table 6), showing that improvements in feasibility and grounding without sacrificing clarity or novelty. Novelty scores for HARPA averaged $5.98 \pm 1.33$ compared to $6.43 \pm 1.32$ for the baseline, with both systems rarely falling below the midpoint of the 10-point scale. This indicates that HARPA produces ideas perceived as incrementally novel. This difference is not statistically significant, and both systems fall within the same *incremental-to-reasonably-novel* range defined in the evaluation rubric. These findings align with our design goal that grounding research proposals in literature and refining

---

[3]Since source papers were selected individually, proposals were unique to each expert and not cross-reviewed.

[4]`claude-sonnet-4` was used as an Oracle model.

hypotheses through a human-like workflow leads to more operational, testable research proposals. (Detailed rating distributions in Appendix A.1.)

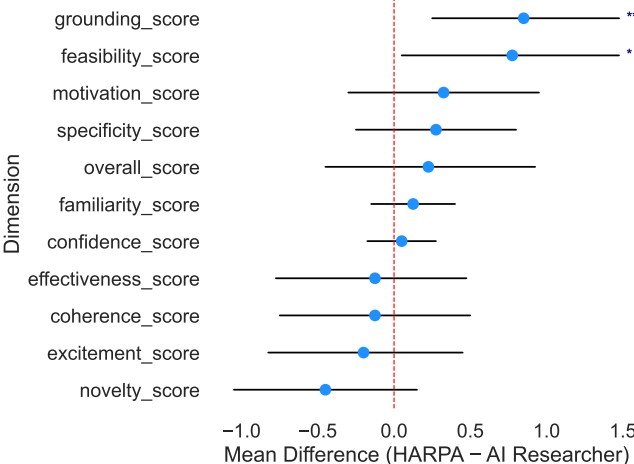

Figure 4: Mean difference between HARPA's proposal generator and AI-Researcher across nine evaluation dimensions. Also reporting the familiarity and confidence score differences. Points show average differences, horizontal bars indicate 95% bootstrap confidence intervals ($10k$ resamples). Stars indicate significant difference computed using the nonparametric bootstrap test (* $p < 0.05$, ** $p < 0.01$)

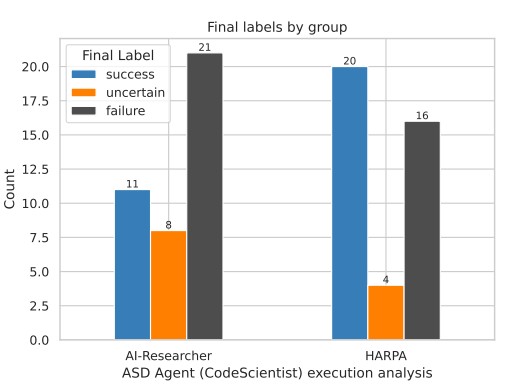

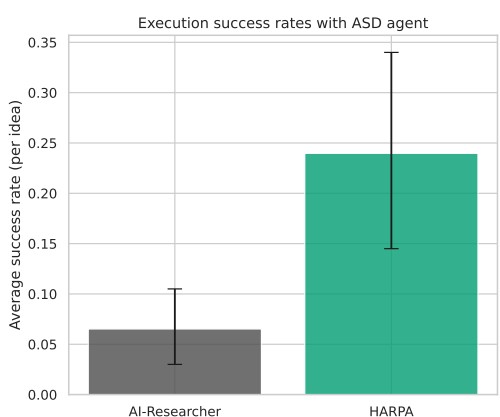

(a) Execution outcomes (counts) from CODESCIENTIST runs, labeled as success, uncertain, or failure.

(b) Average per-idea success rates (95% bootstrap CIs) for HARPA vs. baseline AI-Researcher proposals.

Figure 5: Execution results from CODESCIENTIST. Left: outcome distribution across groups. Right: paired comparison of mean success rates showing HARPA significantly outperforms the baseline AI-Researcher.

**Execution Success rates:** We evaluated whether HARPA proposals more often succeed when executed by a typical ASD agent (here, CODESCIENTIST). For each user-evaluated proposal, we executed five independent runs labeled outcomes using the meta-analysis labeling scheme (Section 3.2). Figure 5a shows the raw distribution of execution outcomes across groups. HARPA produced a higher number of successful executions (20 vs. 11 for the baseline) and fewer outright failures than the baseline AI-researcher system. We further aggregated results into per-idea success rate, defined as the proportion of faithful executions per idea. To ensure fair comparison, we paired HARPA and baseline proposals generated from the same source paper and computed within-source paper differences in success rates. Figure 5b summarizes per-idea success rates where HARPA achieved a higher mean success rate than the baseline (0.24 vs. 0.065), and the bootstrap test ($p < 0.001$) confirmed the difference was statistically significant. These results together demonstrate that HARPA proposals succeed more often in automated executions, consistent with expert ratings of higher feasibility.

| System | Pairwise Consistency | | | | | | Accuracy |
|---|---|---|---|---|---|---|---|
| | Execution Success | Complexity | Cost Efficiency | Hypothesis Validity | Interesting-ness | Faithfulness | (win) |
| Baseline | – | – | – | – | – | – | 0.52 |
| HARPA | 0.80 | 0.67 | 0.73 | 0.69 | 0.55 | 0.79 | **0.81** |

Table 2: Pairwise consistency of **HARPA-Scorer** with oracle judgments across rubrics. Consistency is the fraction of proposal pairs where the scorer and oracle agree. Baseline lacks rubric-level judgments (–). Accuracy comparing baseline and **HARPA-Scorer** on the test data (N=186 success-failure pairs).

| System | BLEU | BLEU BP | ROUGE-1 | ROUGE-2 | ROUGE-L | ROUGE-Lsum | Len-ratio |
|---|---|---|---|---|---|---|---|
| Baseline | 0.08 | 0.79 | 0.43 | 0.13 | 0.18 | 0.17 | 0.81 |
| HARPA | 0.22 | 1.00 | 0.55 | 0.22 | 0.26 | 0.26 | 1.12 |

Table 3: Overlap-based evaluation of HARPA reasoning traces w.r.t reference traces.

**Qualitative Examples:** To illustrate these quantitative trends, an expert rated a HARPA proposal as highly feasible and grounded: "*Using the softmax trick allows backpropagation/gradient estimation, it is a well known trick and the implementation is not so complicated...*" (feasibility = 7, grounding = 9). The expert highlighted that the ideas was concrete and testable, with direct support from the prior literature. By contrast, a baseline proposal as "*The proposed method looks feasible. The problem is that it lacks details. Everything related to the method is summarized in 2–3 lines in the 'Proposed methods' without any mathematical language. ...*" was judged infeasible (feasibility = 4) and poorly grounded (grounding = 2). Although the expert noted that it was an exciting impact, they emphasized that the lack of detail and irrelevant literature made the proposal impossible to operationalize. Examples of full proposals and expert assessments are provided in Appendix C

**Novelty–Feasibility Trade-off.** HARPA is designed to filter out logically inconsistent or non-executable ideas, not to suppress high-risk innovation. While such feasibility-oriented filtering can modestly reduce groundbreaking ideas, the observed novelty difference is not statistically significant, and experts judged HARPA's proposals to remain more actionable. This reflects novelty–feasibility trade-off (Guo et al., 2025), where highly novel ideas often become infeasible; within this trade-off, HARPA maintains comparable novelty while achieving substantially higher feasibility and grounding. HARPA therefore targets actionable novelty suitable for ASD agents rather than unconstrained creativity.

🔍 **Takeaway:** *In summary, HARPA bridges the gap between ideation and execution: it generates literature-grounded, feasible, and testable research proposals that succeed **nearly twice as often in ASD execution (20 vs. 11, ≈2×)**, while also outperforming prior systems in expert evaluations.*

## 5.2 Agent-centric Results

We next evaluated HARPA-scorer against a baseline untrained LLM scorer, a `Qwen-7B-instruct` (section 4.1). HARPA-scorer improves accuracy with a +0.28 absolute gain (a 53% relative improvement), with more balanced performance across classes. This improvement is largely driven by distilling structured reasoning traces, which provide feasibility cues that the untrained LLM does not capture.

Importantly, scorer inference is far cheaper than executing a full CodeScientist run, so accurate feasibility prediction directly reduces compute by avoiding many failed executions. In our evaluation split, only 45 of 120 proposals succeed end-to-end; with 81% accuracy, the scorer correctly flags most low-feasibility proposals before execution. We further tested 35 proposal pairs from the human-evaluation corpus, which includes a wider mix of computational topics; the scorer attained 74% accuracy. While this does not establish cross-domain generalization, it suggests the scorer can handle some proposals outside the ACL-derived distribution. The scorer is general in design and can be extended to new domains or ASD agents given corresponding execution traces.

In addition, HARPA-scorer produces rubric-aligned reasoning traces with explicit scoring on feasibility, cost efficiency, and complexity, like the teacher model. In contrast, the baseline model produced unstructured free text (e.g., in Appendix L 6) that lacks actionable justifications and 4.84% unknown

predictions. This alignment with oracle-style reasoning makes HARPA-scorer's judgement easier to interpret and more reliable for refining the research proposals.

Beyond accuracy, we further assessed *pairwise consistency*, whether the scorer agrees with the Oracle on which two proposals are preferred for each rubric dimension. HARPA-scorer achieves strong alignment on testability-oriented rubrics, with $0.80$ consistency on *Execution Success* and $0.70$ on *faithfulness*, and moderate alignment on *Complexity, Cost Efficiency, and Hypothesis Validity*, while alignment drops to $0.55$ for the more subjective *Interestingness* dimension. These results indicate that the scorer is capturing reliably testability-related signals while remaining less consistent on subjective criteria. Finally, we compare HARPA-generated rationales with baseline ones. HARPA significantly ($p < 0.01$, paired t-test) outperforms the baseline across all overlap metrics with reference rationales. We see major improvements in BLEU scores (+0.14, a 166% increase) and strong gains in ROUGE-1 (+0.12, +27%), ROUGE-2 (+0.10, +77%), and ROUGE-L/Lsum (+0.09, +49%). The particularly strong improvements in higher-order n-grams—like ROUGE-2 and BLEU's 3-4-gram scores—suggest that HARPA is not just matching individual words better, but is actually producing more coherent text with better content flow and sequencing.

🔍 **Takeaway:** *In summary, HARPA's scorer delivers **+0.28 absolute (∼53% relative) higher accuracy** than an untrained LLM scorer, while providing interpretable rubric-style judgments that enable reliable, execution-informed filtering of research proposals.*

## 6 CONCLUSION AND FUTURE WORK

We presented HARPA, a literature-grounded, testability-driven framework for the open-ended task of hypothesis generation. HARPA systematically extracts research trends, explores existing hypothesis spaces, and converges on testable hypothesis-driven proposals. We introduced an interpretable reward-trained scorer that adapts feasibility judgments to ASD agent capabilities, enabling HARPA to prioritize hypotheses that are executable. Our evaluations show significant improvements in feasibility and grounding, with HARPA's proposals also succeeding more often in automated execution. As the scorer serves as a proxy for resource-intensive experimentation, HARPA enables execution-derived feedback into future proposal generation, selectively refining hypotheses in line with ASD agent capabilities. To our knowledge, HARPA is the first ideation framework to learn directly from execution outcomes, enabling feasibility-aware hypotheses generation, and points to further possible improvements using even richer training data and execution environments. Together, these contributions represent a step toward more capable hypothesis generation tools and help advance the rapidly growing field of AI-driven scientific discovery.

### ETHICS STATEMENT

We honor the Code of Ethics. No personally identifiable information is used in this work. The human evaluators were hired from Upwork using a detailed job post. We had Institutional Review Board (IRB) approval for obtaining written consent from our human evaluators. We shared an example task sheet with complete instructions during the recruitment. The evaluators were duly compensated based on minimum wage in the respective countries and always above their quotation.

### THE USE OF LARGE LANGUAGE MODELS (LLMS)

We used AI-based tools (Claude, ChatGPT, and Grammarly) for lightly polishing the grammar, clarity, and identifying errors, and generating code for plots; all ideas and content are the authors' own.

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

Table 4: Distribution of source papers across shared topics.

| shared_topic | Count |
|---|---|
| Graph Neural Networks and Graph Learning | 3 |
| Generative Models for Images | 2 |
| Recommender Systems with LLMs | 2 |
| Medical and Healthcare Applications | 2 |
| Differentiable Optimization | 2 |
| Bias and Fairness in NLP | 2 |
| Hallucination and Factuality in LLMs | 2 |
| Reinforcement Learning for Scheduling | 2 |
| Retrieval-Augmented Generation and Information Refinement | 1 |
| Continual Learning and Knowledge Distillation | 1 |
| Fake News Detection | 1 |

## A  APPENDIX

### A.1  HUMAN EVALUATION DETAILS

We recruited 12 experts with demonstrated research experience, spanning diverse academic and research backgrounds. The expert pool included 6 PhD students in Computer Science or related fields, 3 faculty members/academic researchers, and 2 postdoctoral researchers. Their expertise covered a broad range of topics in ML and NLP: bias and fairness in language models (4), multilingual and low-resource NLP (3), factuality and hallucination detection (3), code generation and programming with LLMs (2), uncertainty estimation and interpretability (2), and mathematical reasoning/structured predictions (2). Additional specialized domains included recommender systems and IR, mathematical modeling, deep reinforcement learning, and AI safety/robustness. Note that the counts are not mutually exclusive, as evaluators could select multiple primary research areas.

In terms of research experience, half of the participants (6/12) reported 3-5 years of active work in their field, three reported 6-10 years, and three reported 10+ years. As part of the screening, each expert shared their Google Scholar profile (or equivalent evidence of publications). The citation count of experts' scholarly work ranged from 7 to 1256 (median =147, mean=297.3). This distribution shows that our evaluation pool included both early-career researchers and more senior researchers with substantial publication records.

### A.1.1  SOURCE PAPER DOMAINS AND TOPIC DISTRIBUTION

Table 4 summarizes the distribution of source papers across shared topics, obtained by classifying abstracts into broad topics using the same approach from Listing 5. The topics span from graph neural networks to health-care applications, optimization to fairness. Figure 6 and Table 5 summarize the source papers selected by experts. These papers span recent years (2018-2023), show moderate citation counts, and cover diverse venues.

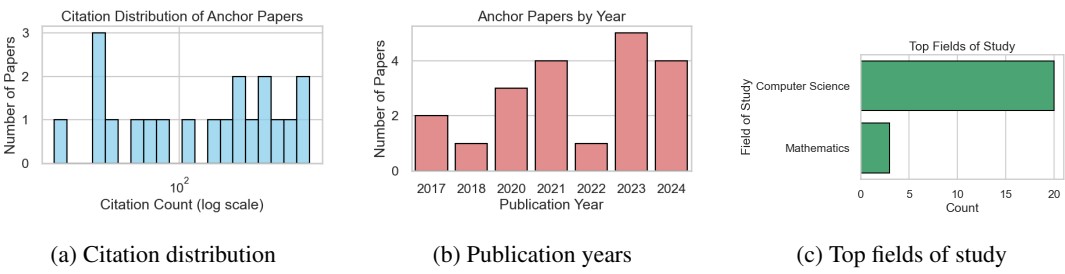

| (a) Citation distribution | (b) Publication years | (c) Top fields of study |
|---|---|---|

Figure 6: Aggregate statistics of source papers selected by experts.

| Venue | Count |
|---|---|
| arXiv.org | 3 |
| Neural Information Processing Systems | 2 |
| Annual Meeting of the Association for Computational Linguistics | 2 |
| Computer Vision and Pattern Recognition | 2 |
| North American Chapter of the Association for Computational Linguistics | 2 |
| International Conference on Computer Graphics and Interactive Techniques | 1 |
| IEEE Access | 1 |
| Knowledge Discovery and Data Mining | 1 |
| ACM Transactions on Intelligent Systems and Technology | 1 |
| ACM Conference on Health, Inference, and Learning | 1 |

Table 5: Venues of source papers selected by experts.

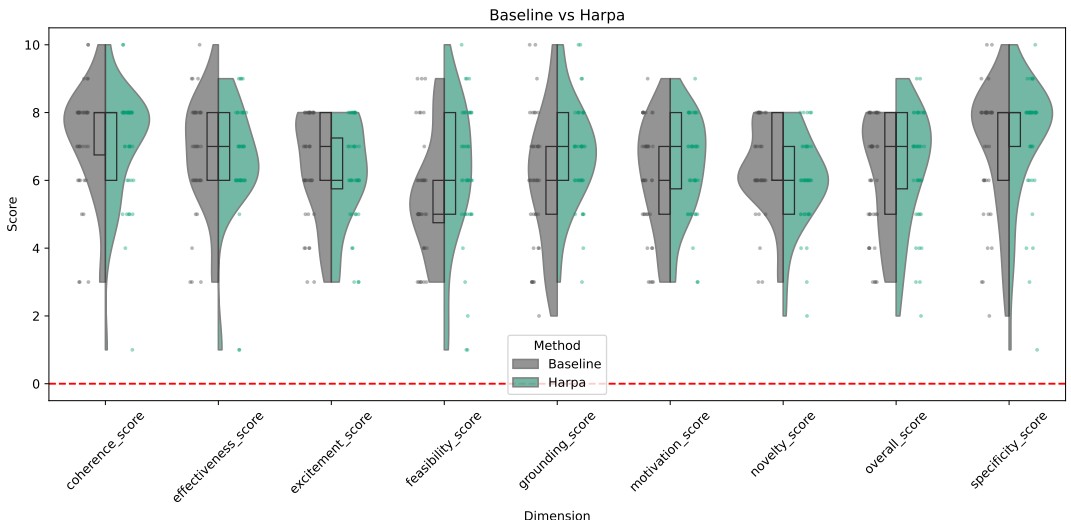

Figure 7: Distribution of expert ratings across nine dimensions for HARPA vs baseline. Shown for completeness (complementary to Fig. 4).

## B  PROPOSAL REVIEW FORM

We use the following proposal assessment form to elicit reviews from all the experts. Our assessment questions largely follow the expert evaluation protocol introduced by Si et al. (2024) for proposal assessment, but we extended it with several additional questions tailored to hypothesis-driven research proposals. In particular, we added dimensions for *motivation*, *specificity*, *coherence*, and *literature grounding*, as these aspects are critical for ensuring that proposals are both operational and directly testable. To ensure consistency, each question was accompanied by a detailed description of the scale points as well as hints on what evidence to consider (e.g., explicit references, prior knowledge).

The full questionnaire (including all Likert-scale anchors and instructions shown to experts) is reproduced below.

**1. Familiarity**: Before reviewing the idea, please indicate how familiar you are with the given topic on a scale of 1 - 5 (this is just for us to understand potential confounders).

1. You have never read about this topic before

2. You have read at least one paper on this topic

3. You have read multiple papers on this topic but have not published any paper on it

4. You have co-authored at least one paper on this topic

5. You have co-authored multiple papers on this topic or have published at least one first-author paper on this topic

**2. Novelty Score**: Whether the idea is creative and different from existing works on the topic, and brings fresh insights. You are encouraged to search for related works online. You should consider all papers that appeared online prior to July 2024 as existing work when judging the novelty.

1. Not novel at all - there are many existing ideas that are the same

2.

3. Mostly not novel - you can find very similar ideas

4.

5. Somewhat novel - there are differences from existing ideas but not enough to turn into a new paper

6. Reasonably novel - there are some notable differences from existing ideas and probably enough to turn into a new paper

7.

8. Clearly novel - major differences from all existing ideas

9.

10. Very novel - very different from all existing ideas in a very interesting and clever way

**Novelty Rationale**: Short justification for your score. If you give a low score, you should specify similar related works. (Your rationale should be at least 2-3 sentences.) *Hint: If the idea is not novel, point out what is already similar in prior work (e.g., method, task, or setting), and briefly mention any minor differences if they exist. If the idea is novel, explain what is new—such as a novel method, a new task, or applying an existing idea to a new domain.*

**3. Feasibility Score**: How feasible it is to implement and execute this idea as a research project? Specifically, how feasible the idea is for a typical CS PhD student to execute within 1-2 months of time. You can assume that we have abundant OpenAI / Anthropic API access, but limited GPU compute.

1. Impossible: the idea doesn't make sense or the proposed experiments are flawed and cannot be implemented

2.

3. Very challenging: there are flaws in the proposed method or experiments, or the experiments require compute/human resources beyond any academic lab

4.

5. Moderately feasible: It can probably be executed within the given time frame but would require careful planning, efficient use of APIs or some advanced computational strategies to overcome the limited GPU resources, and would require some modifications to the original proposal to make it work

6. Feasible: Can be executed within the given constraints with some reasonable planning

7.

8. Highly Feasible: Straightforward to implement the idea and run all the experiments

9.

10. Easy: The whole proposed project can be quickly executed within a few days without requiring advanced technical skills

**Feasibility Rationale**: Short justification for your score. If you give a low score, you should specify what parts are difficult to execute and why. (Your rationale should be at least 2-3 sentences.)

**4. Expected Effectiveness Score**: How likely the proposed idea is going to work well (e.g., better than existing baselines).

1. Extremely Unlikely: The idea has major flaws and definitely won't work well

2.

3. Low Effectiveness: The idea might work in some special scenarios but you don't expect it to work in general

4.

5. Somewhat ineffective: There might be some chance that the proposed idea can work better than existing baselines but the improvement will be marginal or inconsistent

6. Somewhat effective: There is a decent chance that the proposed idea can beat existing baselines by moderate margins on a few benchmarks

7.

8. Probably Effective: The idea should offer some significant improvement over current methods on the relevant benchmarks

9.

10. Definitely Effective: You are very confident that the proposed idea will outperform existing methods by significant margins on many benchmarks

**Expected Effectiveness Rationale**: Short justification for your score. (Your rationale should be at least 2-3 sentences.) *Hint: You must consider how the novelty of the idea relates to its excitement or impact — if the idea is not novel (e.g., already done before), it should generally not be rated as very exciting.*

**5. Excitement Score**: How exciting and impactful this idea would be if executed as a full project. Would the idea change the field and be very influential.

1. Poor: You cannot identify the contributions of this idea, or it's not interesting at all and you would fight to have it rejected at any major AI conference

2.

3. Mediocre: this idea makes marginal contributions and is very incremental

4.

5. Leaning negative: it has interesting bits but overall not exciting enough

6. Learning positive: exciting enough to be accepted at a major AI conference, but still has some weaknesses or somewhat incremental

7.

8. Exciting: would deepen the community's understanding or make major progress in this research direction

9.

10. Transformative: would change the research field profoundly and worth a best paper award at major AI conferences

**Excitement Rationale**: Short justification for your score. (Your rationale should be at least 2-3 sentences.) *Hint: You must consider how the novelty of the idea relates to its excitement or impact — if the idea is not novel (e.g., already done before), it should generally not be rated as very exciting.*

**6. Literature Grounding**: To what extent are the key components (e.g., model choice, tasks, evaluation strategies) grounded in existing scientific literature? You may also consider whether ideas reflect well-established domain knowledge or listed references.

1. Not at all grounded: Mostly speculative or hallucinated; no support from literature or well-established concepts

2.

3. Weak grounding: A few connections to existing work, but most claims lack clear support from the listed references or alignment with well-known concepts

4.

5. Partially grounded: About half the components are linked to literature or reflect widely accepted ideas in the field

6. Strong grounding: Most core elements are supported by the listed references or well-established concepts, with only minor gaps

7.

8. Very strong grounding: The vast majority of components are supported by listed references or widely accepted domain knowledge, though one or two key claims still lack clear support

9.

10. Fully grounded: Every major concept and step is well-supported by listed references or clearly based on well-established domain knowledge; no unsupported claims remain.

**Explanation**: You should also provide a rationale for your score. (Your rationale should be at least 2-3 sentences.) *Hint: If a claim is grounded in well-known concepts but not supported by the listed references, explain why it is reasonable based on your domain knowledge. Indicate whether your assessment relies on (a) the proposal's reference list, (b) external sources you know, or (c) generally accepted field knowledge.*

**7. Motivation from Literature**: Is the problem statement/overall idea clearly defined and motivated by a specific, well-scoped research gap, or limitation identified in the widely recognized field knowledge?

1. No clear motivation: idea feels arbitrary or disconnected

2.

3. Weakly motivated: mentions general themes but lacks a compelling rationale

4.

5. Somewhat motivated: a recognizable problem is present, but vague

6. Well motivated: builds on a clear and relevant research direction

7.

8. Strongly motivated: clearly addresses a known issue or opportunity from existing work or widely acknowledged field challenges

9.

10. Exceptionally motivated: makes a compelling case for a timely and important problem grounded in the reference list or broadly recognized research needs

**Explanation**: You should also provide a rationale for your score. (Your rationale should be at least 2-3 sentences.) Also specify which part of the idea was most clearly linked to a literature-based motivation.

**8. Coherence of Idea Composition**: Are the combined components (problem, methods, tasks, and metrics) logically integrated and literature-informed?

1. Incoherent: parts don't fit together; lacks logical or conceptual connection

2.

3. Loosely connected: some rationale exists, but combination feels forced

4.

5. Reasonable fit: elements are compatible, though not deeply integrated

6. Moderate coherence: combination makes general sense with limited justification

7.

8. Coherent and justified: combination makes sense and is literature-informed

9.

10. Highly coherent: seamless integration of ideas with strong literature basis

**Explanation**: You should also provide a rationale for your score. (Your rationale should be at least 2-3 sentences.) *Hint: If the fit between components is strong, note which elements are well connected and clearly defined for implementation (e.g., problem-task pairing, method-metric match). If weak, specify which parts feel vague, disconnected, or hard to execute.*

**9. Specificity of Proposed Method**: How clearly does the proposed method present a testable research goal or hypothesis? To what extent is it sufficiently detailed to be operationalized in a way that aligns with prior literature or accepted practices?

1. Extremely unclear: the method is explained in an extremely vague or ambiguous manner, making it impossible to understand or replicate the approach without additional information or clarification.

2.

3. Unclear: the method is described with some detail, but significant gaps in explanation or logic leave the reader with considerable confusion and uncertainty about how to apply or replicate the approach.

4.

5. Somewhat clear: method is described with sufficient detail to understand the basic approach, but important elements remain vague or underdeveloped

6. Moderately clear: method is described with sufficient detail to understand the basic approach, but lacks the precision or specificity needed to fully replicate or grasp the nuances of the methodology without further guidance.

7.

8. Clear and testable: method is clearly and precisely described, with most details provided to allow for replication and comprehension, though minor areas may benefit from further clarification or elaboration.

9.

10. Highly clear and specific: method is articulated in an exceptionally clear, precise, and detailed manner, enabling straightforward replication and thorough understanding of the approach with no ambiguities

**Explanation**: You should also provide a rationale for your score. (Your rationale should be at least 2-3 sentences.)

**10. Overall Score**: Overall score: Apart from the above, you should also give an overall score for the idea on a scale of 1 - 10 as defined below (Major AI conferences in the descriptions below refer to top-tier NLP/AI conferences such as *ACL, COLM, NeurIPS, ICLR, and ICML.):

1. Critically flawed, trivial, or wrong, would be a waste of students' time to work on it

2. Strong rejection for major AI conferences

3. Clear rejection for major AI conferences

4. Ok but not good enough, rejection for major AI conferences

5. Decent idea but has some weaknesses or not exciting enough, marginally below the acceptance threshold of major AI conferences

6. Marginally above the acceptance threshold of major AI conferences

7. Good idea, would be accepted by major AI conferences

8. Top 50% of all published ideas on this topic at major AI conferences, clear accept

9. Top 15% of all published ideas on this topic at major AI conferences, strong accept

10. Top 5% of all published ideas on this topic at major AI conferences, will be a seminal paper

**Overall Rationale**: You should also provide a rationale for your overall score. (Your rationale should be at least 2-3 sentences.) *Hint: This is just an idea. Please evaluate its potential — assuming it is properly fleshed out, implemented, and empirically validated, would it be acceptable at a future major AI conference? If the idea is too vague to envision as a strong paper, it should be rated lower.*

**11. Confidence**: Additionally, we ask for your confidence in your review on a scale of 1 to 5 defined as following:

1. Your evaluation is an educated guess

2. You are willing to defend the evaluation, but it is quite likely that you did not understand central parts of the paper

3. You are fairly confident that the evaluation is correct

4. You are confident but not absolutely certain that the evaluation is correct

5. You are absolutely certain that the evaluation is correct and very familiar with the relevant literature

**Time**: How many minutes did you spend on this task?

| Dimension | Baseline | | | | HARPA | | | |
|---|---|---|---|---|---|---|---|---|
| | Mean | Std | Min–Max | Median | Mean | Std | Min–Max | Median |
| Coherence | 7.20 | 1.65 | 3–10 | 8.0 | 7.08 | 1.65 | 1–10 | 8.0 |
| Confidence | 4.33 | 0.66 | 3–5 | 4.0 | 4.38 | 0.67 | 3–5 | 4.0 |
| Effectiveness | 6.78 | 1.66 | 3–10 | 7.0 | 6.65 | 1.69 | 1–9 | 7.0 |
| Excitement | 6.45 | 1.54 | 3–8 | 7.0 | 6.25 | 1.48 | 3–8 | 6.0 |
| Familiarity | 3.93 | 0.94 | 2–5 | 4.0 | 4.05 | 0.93 | 2–5 | 4.0 |
| Feasibility | 5.50 | 1.72 | 3–9 | 5.0 | 6.28 | 2.08 | 1–10 | 6.0 |
| Grounding | 5.98 | 1.94 | 2–10 | 6.0 | 6.83 | 1.47 | 3–10 | 7.0 |
| Motivation | 6.13 | 1.64 | 3–9 | 6.0 | 6.45 | 1.43 | 3–9 | 7.0 |
| Novelty | 6.43 | 1.32 | 3–8 | 6.0 | 5.98 | 1.33 | 2–8 | 6.0 |
| Overall | 6.20 | 1.71 | 3–8 | 7.0 | 6.43 | 1.69 | 2–9 | 7.0 |
| Specificity | 7.00 | 1.88 | 2–10 | 8.0 | 7.28 | 1.78 | 1–10 | 8.0 |

Table 6: Expert ratings across 11 dimensions. Values report mean, std, min–max, and median (10-point Likert scale, higher is better). $n = 40$ proposals per system.

## B.1 Statistical Tests for Human Evaluation

For each dimension, we computed paired differences between HARPA and the baseline on expert ratings. Statistical significance was assessed using bootstrap resampling (10,000 iterations) and Wilcoxon signed-rank tests. We report bootstrap as our primary test, since it makes no distributional assumptions and is appropriate for small sample sizes and ordinal scores. Table 7 shows the mean differences and $p$-values.

| Dimension | MeanDiff | Bootstrap_p | Boot* | Wilcoxon_p | Wilcoxon* |
|---|---|---|---|---|---|
| coherence_score | -0.125 | 0.666 | | 0.806 | |
| effectiveness_score | -0.125 | 0.663 | | 0.716 | |
| excitement_score | -0.200 | 0.753 | | 0.435 | |
| familiarity_score | 0.125 | 0.210 | | 0.394 | |
| confidence_score | 0.050 | 0.360 | | 0.660 | |
| feasibility_score | 0.775 | 0.017 | * | 0.016 | * |
| grounding_score | 0.850 | 0.002 | ** | 0.017 | * |
| motivation_score | 0.325 | 0.163 | | 0.286 | |
| novelty_score | -0.450 | 0.937 | | 0.107 | |
| overall_score | 0.225 | 0.275 | | 0.598 | |
| specificity_score | 0.275 | 0.168 | | 0.430 | |

Table 7: Mean differences (HARPA – baseline) with significance tests. Stars indicate significance (* $p < 0.05$, ** $p < 0.01$). Bootstrap resampling is our primary test.

## C Full examples of expert review and proposals

Table 8 shows two representative pairs of hypotheses (HARPA vs. baseline), along with expert assessment across all evaluation dimensions. Each row corresponds to one proposal. Complete dataset generated and assessed for human evaluation are available in the supplementary files.

We include an example of full proposal evaluated by experts and generated by HARPA and by the baseline ideator.

HARPA PROPOSAL

# Paper ID

3bfb5f836d944414c171f8f843eaf90cf5604243

---

# Title

Combining stochastic softmax tricks with control variates for improved spanning tree optimization.

---

# Introduction

## Problem Statement

Integrating stochastic softmax tricks with control variates will significantly improve convergence speed and stability in spanning tree optimization problems compared to using stochastic softmax tricks alone.

## Motivation

Existing methods for variance reduction in discrete optimization problems often focus on individual techniques like Rao-Blackwellization or stochastic softmax tricks in isolation. However, these approaches do not fully exploit the potential synergies between different variance reduction techniques, particularly in complex combinatorial spaces like spanning trees and arborescences. No prior work has explored the integration of stochastic softmax tricks with control variates specifically for spanning tree problems, which could offer significant improvements in convergence speed and stability by leveraging structured relaxations and variance reduction simultaneously.

---

# Proposed Method

The research aims to explore the integration of stochastic softmax tricks with control variates to enhance variance reduction in spanning tree optimization problems. Stochastic softmax tricks provide structured relaxations that allow for gradient estimation in combinatorial spaces, while control variates reduce the variance of these gradient estimators by incorporating additional information. By combining these two techniques, the hypothesis posits that the model will achieve faster convergence and more stable performance. This approach addresses the gap in existing research where these techniques are typically applied in isolation. The expected outcome is a more efficient optimization process, particularly in graph-based problems like spanning trees, where maintaining the graph structure is crucial for accurate gradient computation. This combination is expected to reduce the variance of gradient estimates more effectively than either technique alone, leading to improved model performance metrics such as convergence speed and stability.

## Background

Stochastic Softmax Tricks: Stochastic softmax tricks are used to create structured relaxations for combinatorial optimization problems, such as spanning trees. This involves using the Gumbel-Max trick to reparameterize distributions over one-hot binary vectors, allowing for gradient estimation in discrete distributions. The structured relaxation maintains the graph structure, enabling efficient gradient computation. This technique was selected for its ability to handle complex combinatorial spaces and its compatibility with gradient-based optimization methods.

Control Variates: Control variates are used to reduce the variance of gradient estimators by incorporating additional information into the estimation process. This involves constructing a control variate based on an analytical linear approximation to the gradient estimator, which is then combined with a naïve gradient estimate. This method remains unbiased while achieving lower variance, particularly effective in Gaussian approximating families. The control variate is expected to enhance the efficiency of the stochastic softmax tricks by further reducing the variance of the gradient estimates.

## Implementation

The proposed method involves integrating stochastic softmax tricks with control variates to optimize spanning tree problems. First, the stochastic softmax tricks are applied to

create a structured relaxation of the spanning tree problem, allowing for gradient estimation in a differentiable manner. This is achieved by representing the problem as a linear program and applying a softmax function to approximate the selection of edges. Next, control variates are introduced to further reduce the variance of the gradient estimators. This involves constructing a control variate based on an analytical linear approximation to the gradient estimator, which is then combined with the gradient estimates obtained from the stochastic softmax tricks. The integration occurs at the gradient computation stage, where the control variate is used to adjust the gradient estimates, leading to lower variance and improved convergence. The data flows from the structured relaxation to the control variate adjustment, with the final output being a more stable and efficient gradient estimate. This method is implemented using libraries that support automatic differentiation, such as TensorFlow or PyTorch, and is evaluated against baseline methods like the vanilla Gumbel-Softmax estimator.

---

# Experiments Plan

## Operationalization Information

Please implement an experiment to test the hypothesis that integrating stochastic softmax tricks with control variates will significantly improve convergence speed and stability in spanning tree optimization problems compared to using stochastic softmax tricks alone.

**Experiment Overview**

This experiment will compare three methods for spanning tree optimization:
1. **Baseline 1**: Vanilla Gumbel-Softmax estimator
2. **Baseline 2**: Stochastic softmax tricks without control variates
3. **Experimental**: Stochastic softmax tricks integrated with control variates

**2. Stochastic Softmax Tricks without Control Variates (Baseline 2)**

Implement structured relaxations for spanning trees:
- Represent the spanning tree polytope using the cycle constraints
- Apply stochastic softmax tricks to maintain the graph structure
- Use automatic differentiation to compute gradients
- Implement a projection step to ensure the solution is a valid spanning tree

**3. Integrated Approach with Control Variates (Experimental)**

The experiment should measure convergence speed (iterations to reach a predefined accuracy threshold) and stability (variance of predictions across different runs) for each method.

**Implementation Details**

**Pilot Mode Settings**

Implement a global variable `PILOT_MODE` with three possible settings: `MINI_PILOT`, `PILOT`, or `FULL_EXPERIMENT`.
- **MINI_PILOT**: Use 5 small random graphs (10-15 nodes) and run 10 optimization iterations with 3 independent runs per method
- **PILOT**: Use 20 medium-sized random graphs (20-50 nodes) and run 50 optimization iterations with 10 independent runs per method
- **FULL_EXPERIMENT**: Use 100 graphs of varying sizes (up to 100 nodes) and run 200 optimization iterations with 30 independent runs per method

The experiment should first run in MINI_PILOT mode, then PILOT mode if successful, but stop before FULL_EXPERIMENT (which will be manually triggered after human verification).

**Graph Dataset Generation**

Use NetworkX to generate the following types of random graphs for the experiment:
1. Erdős–Rényi random graphs

Extend the stochastic softmax tricks implementation with control variates:
- Construct a control variate based on an analytical linear approximation to the gradient estimator
- Combine the control variate with the naïve gradient estimate from the stochastic softmax tricks
- Implement the optimal scaling parameter for the control variate
- Apply the adjusted gradient in the optimization process

**Optimization Task**

Implement a minimum spanning tree optimization task where the objective is to find the spanning tree with minimum total edge weight. Additionally, implement a maximum spanning tree task as a secondary objective.

**Evaluation Metrics**

1. **Convergence Speed**: Measure the number of iterations required to reach 95% of the optimal solution
2. **Stability**: Calculate the variance of the solutions across multiple independent runs
3. **Solution Quality**: Compare the final solution to the true optimal spanning tree (computed using standard MST algorithms)
4. **Gradient Variance**: Measure the variance of the gradient estimates during optimization

**Experiment Procedure**

2. Barabási–Albert preferential attachment graphs

3. Watts–Strogatz small-world graphs

For each graph, assign random edge weights from a uniform distribution [0.1, 10.0].

**Method Implementations**

**1. Vanilla Gumbel-Softmax Estimator (Baseline 1)**

Implement the standard Gumbel-Softmax trick for spanning tree optimization:
- Represent the graph as an edge selection problem
- Apply the Gumbel-Max trick to sample spanning trees
- Use the softmax temperature parameter to control the discreteness of the distribution
- Implement straight-through estimation for the backward pass

1. For each graph in the dataset:
   a. Run each method (Baseline 1, Baseline 2, Experimental) multiple times with different random seeds
   b. Record the optimization trajectory (objective value vs. iteration)
   c. Measure the gradient variance at each iteration
   d. Calculate the final solution quality

1. Aggregate results across all graphs and runs:
   a. Calculate average convergence speed for each method
   b. Calculate average stability for each method
   c. Perform statistical significance tests (bootstrap resampling) to compare methods

**Visualization and Reporting**

1. Generate convergence plots showing objective value vs. iteration for each method
2. Create box plots showing the distribution of convergence speeds and stability metrics
3. Generate tables with summary statistics for each method
4. Visualize example spanning trees produced by each method on selected graphs

**Implementation Notes**

1430
1431
1432
1433
1434
1435
1436
1437
1438
1439
1440
1441
1442
1443
1444
1445
1446
1447
1448
1449
1450
1451
1452
1453
1454
1455
1456
1457
1458
1459
1460
1461
1462
1463
1464
1465
1466
1467
1468
1469
1470
1471
1472
1473
1474
1475
1476
1477
1478
1479
1480
1481
1482
1483
1494

- Use PyTorch for automatic differentiation and gradient computation
- Use NetworkX for graph manipulation and visualization
- Implement proper seeding for reproducibility
- Use a learning rate scheduler to improve convergence
- Save intermediate results to allow for experiment resumption

**Expected Output**

The experiment should produce:
1. A comprehensive report comparing the three methods
2. Convergence plots for each method
3. Statistical analysis of the differences between methods
4. Visualizations of example spanning trees
5. Raw data for further analysis

Please implement this experiment and run it first in MINI_PILOT mode, then in PILOT mode if successful. Do not proceed to FULL_EXPERIMENT mode without human verification.

*End Note:*

*The source paper is* Paper 0: Learning with Differentiable Perturbed Optimizers *(109 citations, 2020). This idea draws upon a trajectory of prior work, as seen in the following sequence:* Paper 1 --> Paper 2 --> Paper 3 --> Paper 4 --> Paper 5*. The analysis reveals a consistent theme of addressing the high variance in gradient estimation for discrete latent variables, a challenge initially highlighted in the source paper. The*

*progression of research has introduced various techniques like stochastic softmax tricks, Rao-Blackwellization, and coupled gradient estimators to tackle this issue. However, these approaches often focus on specific applications or settings, such as combinatorial spaces or categorical variables. A novel research idea could involve developing a generalized framework that unifies these variance reduction techniques, making them adaptable to a broader range of discrete optimization problems. This would advance the field by providing a more versatile tool for training models with discrete components, addressing the limitations of existing methods that are often application-specific.*

*The initial trend observed from the progression of related work highlights a consistent research focus. However, the final hypothesis proposed here is not merely a continuation of that trend — it is the result of a deeper analysis of the hypothesis space. By identifying underlying gaps and reasoning through the connections between works, the idea builds on, but meaningfully diverges from, prior directions to address a more specific challenge.*

---

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

BASELINE PROPOSAL

# Paper ID

3bfb5f836d944414c171f8f843eaf90cf5604243

# Title

Adaptive Stochastic Gradient Clipping: Enhancing Stability and Convergence in Deep Learning Pipelines

# Introduction

## Problem Statement

Gradient-based optimization in deep learning often suffers from instability and slow convergence, especially in complex decision-making pipelines where gradients can become extremely large or vanishingly small. This issue can lead to poor model performance, slow training, and difficulties in fine-tuning models for specific tasks.

## Motivation

Existing methods like fixed gradient clipping, adaptive learning rates, and normalization techniques often struggle to balance stability and convergence speed across different layers and tasks within a pipeline. Inspired by the success of noise injection in improving generalization and the adaptive nature of biological neural systems, we propose a method that dynamically adjusts gradient updates based on local statistics and stochastic perturbations. This approach allows for aggressive updates in stable regions while dampening oscillations in sensitive areas, all while introducing beneficial noise for improved exploration and generalization.

# Proposed Method

We introduce Adaptive Stochastic Gradient Clipping (ASGC), which combines layer-wise gradient statistics with controlled stochastic perturbations. For each layer, we maintain running estimates of gradient mean and variance. During each update, we compute a clipping threshold as a function of these statistics. Before applying the threshold, we add Gaussian noise scaled by the layer's gradient variance. The clipping function is smoothed using a differentiable approximation, allowing end-to-end training. The noise scale and clipping function parameters are meta-learned across a diverse set of tasks.

---

# Experiments Plan

## Step-by-Step Experiment Plan

### Step 1: Implement ASGC

Implement the ASGC algorithm as a PyTorch optimizer. This involves creating a custom optimizer class that inherits from torch.optim.Optimizer and overrides the step() method. The key components are: (1) Maintaining running estimates of gradient mean and variance for each layer. (2) Computing the adaptive clipping threshold. (3) Adding scaled Gaussian noise to the gradients. (4) Applying the smoothed clipping function. (5) Updating the parameters using the clipped and noisy gradients.

### Step 2: Prepare Datasets

Prepare the following datasets for evaluation: (1) ImageNet for image classification. (2) WMT14 English-German for machine translation. (3) Atari suite (specifically Breakout, Pong, and Space Invaders) for reinforcement learning.

### Step 3: Setup Baseline Models

Implement baseline models for each task: (1) ResNet-50 for ImageNet. (2) Transformer for WMT14. (3) DQN for Atari games. Train these models using standard optimizers: Adam, SGD with momentum, Adagrad, and RMSprop.

**Step 4: Train Models with ASGC**

Train the same model architectures using ASGC. Use a grid search to find optimal hyperparameters for ASGC, including the initial noise scale and clipping function parameters.

**Step 5: Evaluate Performance**

Compare ASGC against baselines on the following metrics: (1) Final test accuracy/BLEU score/game score. (2) Training time to reach a specific performance threshold. (3) Stability of training (measured by the variance of validation performance across epochs). (4) Generalization (measured by the gap between training and test performance).

**Step 6: Analyze Robustness**

Evaluate the robustness of ASGC to hyperparameter choices by training models with randomly sampled hyperparameters and comparing the distribution of final performances against baselines.

**Step 7: Visualize Gradient Statistics**

Plot the distribution of gradient magnitudes before and after clipping for different layers and at different stages of training. Compare these distributions between ASGC and baseline optimizers.

**Step 8: Analyze Meta-Learned Parameters**

Examine the learned noise scales and clipping function parameters across different tasks and model architectures. Visualize how these parameters evolve during training.

**Step 9: Ablation Studies**

Conduct ablation studies to isolate the effects of adaptive clipping and stochastic perturbations. Train models with only adaptive clipping (no noise) and only stochastic perturbations (fixed clipping threshold).

**Step 10: Write Up Results**

Compile all results, visualizations, and analyses into a comprehensive report or paper draft.

## Test Case Examples

**Baseline Prompt Input**

Train a ResNet-50 model on ImageNet using Adam optimizer with default hyperparameters.

**Baseline Prompt Expected Output**

Final Top-1 Accuracy: 76.1%, Training Time: 90 hours, Stability (std dev of validation accuracy over last 10 epochs): 0.5%

**Proposed Prompt Input**

Train a ResNet-50 model on ImageNet using ASGC optimizer with meta-learned hyperparameters.

**Proposed Prompt Expected Output**

Final Top-1 Accuracy: 77.3%, Training Time: 85 hours, Stability (std dev of validation accuracy over last 10 epochs): 0.3%

**Explanation**

ASGC achieves higher accuracy in less training time, with improved stability during the final stages of training. This demonstrates the benefits of adaptive clipping and stochastic perturbations in balancing aggressive updates and stability.

## Fallback Plan

If ASGC does not outperform baselines as expected, we can pivot the project to an in-depth analysis of why adaptive stochastic methods struggle in certain scenarios. We would conduct a series of experiments to isolate the effects of gradient clipping, noise injection, and adaptive thresholds on different types of neural architectures and tasks. This could involve visualizing gradient flow through networks, analyzing the spectrum of the Hessian at different stages of training, and studying how different optimization techniques affect the loss landscape. We could also explore combining ASGC with other advanced optimization techniques like layer-wise adaptive rates or Hessian-based preconditioning. The goal would be to provide insights into the interplay between network architecture, task complexity, and optimization dynamics, potentially informing the development of next-generation optimization algorithms.

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

| System | Hypothesis/Problem Statement, Ratings, and Justification |
|---|---|
| HARPA | *"Integrating stochastic softmax tricks with control variates will significantly improve convergence speed and stability in spanning tree optimization problems compared to using stochastic softmax tricks alone."* 

 **Novelty = 6** ("The following paper is a neurips 2020 paper that has spanning tree optimization as an application: ""Gradient Estimation with Stochastic Softmax Tricks "". It mainly uses softmax trick for some discrete problems such as spanning tree optimization. The main novelty comes to add the control variates into the loop.") 

 **Feasibility = 7** ("Using the softmax trick allows backpropagation/gradient estimation, it is a well known trick and the implementation is not so complicated, although thee control variates is not so clear how would be implemented (with examples) ") 

 **Grounding = 9** ("It is very grounded on the listed refereneces, mostly similar to "Gradient Estimation with Stochastic Softmax Tricks (2020)". The control variates ideas, although slightly mentioned in the same paper, are more frequent described at "A generalized approximate control variate framework for multifidelity uncertainty quantification (2018)" ") 

 **Specificity = 6** ("I feel the control variates part is not so clear how it would be implemented. What is the additional variable that is correlated to the target? what would be the target in that case? I can see the motivation for that, but it is not so clear how it would be implemented. Examples would be appreciated. ") 

 **Coherence = 8** ("It is very clear that softmax trick is for gradient estimation and control variates is where the novelty is, to improve efficiency. So I see it is coherent. ") 

 **Motivation = 6** ("The motivation only comes from adding control variates to speedup convergence to the already existing methods using "stochastic softmax tricks". I can't see anything else regarding motivation. ") 

 **Excitement = 5** (""I would say it is not exciting due to the lack of novelty (compared to the given neurips paper in 2020). The experiments are also vanilla (mostly ablation studies). The experiment overview is basically removing the control variates and the softmax to compare with the method with both. ") 

 **Effectiveness = 6** ("It is very clear that softmax trick is for gradient estimation and control variates is where the novelty is, to improve efficiency. So I see it is coherent. ") 

 **Overall = 6** ("The motivation of the proposed method is clear, increase efficiency when bringing the control variates into the loop. However, details on how to incorporate the control variate ideas, which is the core of the novelty, are not so clear. ") 

 **Confidence = 4** |
| Baseline | *"Gradient-based optimization in deep learning often suffers from instability and slow convergence, especially in complex decision-making pipelines where gradients can become extremely large or vanishingly small. This issue can lead to poor model performance, slow training, and difficulties in fine-tuning models for specific tasks."* 

 **Novelty = 5** ("From one perspective, I don't score high the novelty regarding this proposal since this would depend on some detais that are not explicitly mentioned. Example: "During each update, we compute a clipping threshold as a function of these statistics". How exactly those statistics will be used would impact on the novelty. For example, Adam optimizer also use statistics for updating gradients. However, the overall method has its novely characteristics by combining the statistics with stochastic perturbations.") 

 **Feasibility = 4** ("The proposed method looks feasible. The problem is that it lacks details. Everything related to the method is summarized in 2 or 3 lines in the "Proposed methods" without any mathematical language. How would we smooth the clipping function, for example?") 

 **Grounding = 2** ("The proposal reference list is not linked to the proposed methods. Soe papers I am really aware of have nothing to do with the method proposed. For example, "Differentiation of Blackbox Combinatorial Solvers (2019)" is specifically about imitation learning of combinatorial labels, there is no novelty towards unconstrained optimizers.") 

 **Specificity = 2** ("As I mentioned before, the details are the problem in this proposal. There is no details of the proposed methods, and therefore the specificity is extremely unclear.") 

 **Coherence = 6** ("The proposed method is very weak and lack of important details. The experimental plan looks all correct, but they are not so important with respect to the method itself. e, it is obvious from the Proposed metod that it is a new optimizer, and then the step 1 of experimental plan it just repeat the steps without detailing it.") 

 **Motivation = 6** ("The section of motivation makes sense, although not grounded in the list of papers.") 

 **Excitement = 7** ("It is definitely exciting in a sense that the impact of this type of research is extremely high, since it can substitute, for example, specific pytorch optimizers that are widely used (for example, Adam) and sometimes suffer from convergences difficulties depending on the architecture used (for example, RNNs).") 

 **Effectiveness = 3** ("The main problem of this research proposal is the lack of details in the proposed method subsection. I still don't see how the parts that matter would be really implemented, such as the clipping part. And what is end-to-end? What is optimized for the adaptation?") 

 **Overall = 3** ("The main problem of this proposal is the lack of details. The method itself, in a high-level idea, makes sense. But the "how" is extremely unclear. There is no depth of the methodology. The ideas ends up in the high-level buzzwords.") 

 **Confidence = 4** |

Table 8: Representative HARPA vs. baseline hypotheses with expert assessment.

# D  HARPA-SCORER

## D.1  HARPA-SCORER: ADDITIONAL EXPERIMENT DETAILS

**Implementation details.**  We follow almost the same hyperparameters as the RM-R1 pipeline (Chen et al., 2025c), consisting of two stages: (i) reasoning distillation from oracle rubric-style traces, and (ii) RLVR fine-tuning on execution-derived preference pairs. The backbone is `Qwen2.5-7B-Instruct`, trained with `openrlhf` and DeepSpeed using full fine-tuning.

**Distillation Stage.**  We fine-tuned on 3,459 rubric-aligned preference pairs (Section 4.3), split into 2,595 train, 452 validation, and 412 test. Training used a global batch size of 4 (micro-batch size 1), maximum sequence length 12,288, and Adam optimizer with offloading at a learning rate of $5 \times 10^{-6}$.

We trained for 1 epoch in `bfloat16` precision with ZeRO stage-2 optimization, gradient checkpointing, FlashAttention, and sample packing. Training was performed on $4\times$ NVIDIA A100-SXM4-80GB.

**RLVR Stage.** We further optimized the distilled model with RLVR on execution-derived success/failure pairs, using the `verl` PPO trainer. Training was performed on $8\times$NVIDIA A100-SXM4-80GB GPUs. We set the rollout batch size to 64, PPO mini-batch size to 16, and micro-batch size per GPU to 4. The learning rate was $1 \times 10^{-6}$ with constant warmup. Maximum input and output lengths were both 8,192 tokens. KL regularization was applied with a coefficient of $10^{-3}$ and a clip ratio of 0.2, with entropy coefficient set to 0. GPU memory utilization was capped at 0.5 to prevent OOM issues. Sampling used temperature 1.0 and top-p 1.0. Training ran for a single epoch, with checkpoints saved every 1,000 steps.

**Compute Cost.** The distillation stage required 3h08m of training on $4\times$A100-80GB GPUs, and the RLVR stage required 8d10h on $8\times$A100-80GB GPUs. Constructing the scorer's training dataset is the dominant cost, since each supervision signal requires a full ASD-agent execution (code synthesis and experiment run), as described in Section 4.3.

---

**Example reasoning trace**

```
<rubric>
 1. Execution Success:
   - Proposal A: 0.9 - Well-structured multilingual toxicity detection experiment
   ...  with clear baseline/experimental comparison, proper statistical analysis, and
   ...  realistic implementation scope
   - Proposal B: 0.7 - Complex RAG system comparison with multiple components that
   ...  may face integration challenges and dataset loading issues

 2. Complexity:
   - Proposal A: 0.3 - Moderate complexity with toxicity classification, adversarial
   ...  generation, and iterative improvement but manageable scope
   - Proposal B: 0.7 - High complexity requiring multiple RAG architectures,
   ...  real-time retrieval, adaptive mechanisms, and complex integration

 3. Cost Efficiency:
   - Proposal A: 0.8 - Efficient use of pre-trained models for toxicity detection
   ...  with limited language scope in pilot mode
   - Proposal B: 0.6 - Multiple model comparisons and complex retrieval operations
   ...  likely to consume significant computational resources

 4. Expected Hypothesis Validity:
   - Proposal A: 0.6 - Clear metrics (compliance ratio, adversarial accuracy drop)
   ...  but dynamic adversarial integration may not necessarily improve performance
   - Proposal B: 0.7 - Well-defined metrics (BLEU/ROUGE scores) with logical
   ...  expectation that adaptive architectures should improve performance

 5. Expected Interestingness:
   - Proposal A: 0.8 - Novel integration of dynamic data collection with adversarial
   ...  evaluation in multilingual safety context
   - Proposal B: 0.7 - Interesting combination of real-time and adaptive retrieval
   ...  but more incremental advancement

 6. Faithfulness:
   - Proposal A: 0.8 - Clear operationalization with realistic simulation of
   ...  human-in-the-loop processes
   - Proposal B: 0.6 - Complex integration requirements may lead to simplified
   ...  implementations that deviate from intended design

<justify>
```

```
 Proposal A scores higher on execution success due to its more straightforward
...  implementation using established toxicity detection methods, while Proposal B's
...  multiple complex components increase failure risk. Proposal A is less complex,
...  focusing on a single domain (toxicity detection) with clear iterative
...  improvement, whereas Proposal B requires implementing multiple RAG architectures
...  with real-time and adaptive components. Cost efficiency favors Proposal A due to
...  its focused scope and efficient use of pre-trained models. Both proposals have
...  reasonable hypothesis validity, but Proposal B's expectation of improvement from
...  adaptive architectures is slightly more logical. Proposal A offers higher
...  interestingness due to its novel approach to multilingual safety alignment.
...  Faithfulness is higher for Proposal A because its human-in-the-loop simulation
...  is more realistic than Proposal B's complex integration requirements.
</justify>
</rubric>

<eval>
 Both proposals address benchmark creation and evaluation but take different
...  approaches.

<summary_A>Proposal A focuses on multilingual toxicity detection, comparing a
...  baseline system using static data against an experimental system that integrates
...  dynamic data collection with adversarial evaluation</summary_A>. <quote_A>"The
...  experiment should compare a baseline system against an experimental system, with
...  appropriate metrics to measure performance differences"</quote_A> and uses
...  <quote_A>"compliance ratios in multilingual toxicity detection tasks and
...  adversarial accuracy drop"</quote_A> as key metrics.

<summary_B>Proposal B targets retrieval-augmented generation systems, comparing
...  three configurations: baseline RAG, real-time RAG, and dynamic RAG that combines
...  both real-time retrieval and adaptive architectures</summary_B>. <quote_B>"This
...  experiment will compare three RAG system configurations: 1. Baseline RAG: A
...  standard RAG system with static retrieval strategies 2. Real-time RAG: A RAG
...  system with real-time information retrieval 3. Dynamic RAG (Experimental): A RAG
...  system that integrates both real-time information retrieval and adaptive
...  retrieval architectures"</quote_B>.

 In terms of implementation complexity, Proposal A presents a more manageable scope.
...  <quote_A>"Use 100 examples per language from 5 languages (English, Spanish,
...  French, German, Chinese) for training and 50 examples per language for testing.
...  Run 5 iterations of the dynamic data collection process"</quote_A> in pilot
...  mode, with clear simulation strategies: <quote_A>"Since actual human annotators
...  aren't available, simulate them by: 1. Creating a function that identifies model
...  errors and generates challenging examples 2. Implementing different 'annotator
...  personas' with varying expertise"</quote_A>.

 Proposal B involves significantly more complex integration requirements.
...  <quote_B>"Implement a module that: - Dynamically accesses and integrates
...  external knowledge bases during generation - Updates retrieval strategies based
...  on the current generation state - Monitors the generation process to identify
...  when new information is needed"</quote_B> and <quote_B>"Implement an
...  architecture that: - Dynamically adjusts retrieval strategies based on task
...  complexity and information needs - Uses a feedback mechanism to evaluate the
...  quality of retrieved documents - Maintains multiple retrieval
...  strategies"</quote_B>.

 Both proposals include appropriate pilot mode configurations, but Proposal A's
...  approach is more straightforward to implement and debug. <summary_A>Proposal A
...  uses established toxicity detection methods with clear metrics and realistic
...  human-in-the-loop simulation</summary_A>, while <summary_B>Proposal B requires
...  coordinating multiple complex components including real-time retrieval, adaptive
...  architectures, and integration mechanisms</summary_B>.
```

```
   The evaluation metrics are well-defined in both cases, but Proposal A's metrics are
... more directly measurable: <quote_A>"Compliance Ratio: Calculate the proportion
... of model outputs that adhere to safety guidelines" and "Adversarial Accuracy
... Drop: Measure the decrease in model performance when exposed to adversarial
... inputs"</quote_A>. Proposal B uses standard NLG metrics: <quote_B>"BLEU score
... between generated text and reference text" and "ROUGE scores (ROUGE-1, ROUGE-2,
... ROUGE-L)"</quote_B>.

</eval>
<answer>[[A]]</answer>
```

Listing 1: Example of rubric-style reasoning trace given the proposal pairs and their execution metadata from an oracle model

**Example extracted execution trace**

```
{
"summary_results":
{"summary": "This experiment tested whether dynamic sparsification tailored to
...  syntactic complexity of queries and semantic density of documents would improve
...  retrieval speed and accuracy compared to static sparsification methods on the
...  MSMARCO dataset. The experiment used 200 queries with 730 documents in pilot
...  mode. The baseline method employed static TF-IDF with BM25 scoring using a fixed
...  1000 features, while the experimental method dynamically adjusted the number of
...  features (500-1500) based on query complexity scores and applied density-aware
...  similarity weighting. However, the results showed identical performance between
...  both methods: both achieved exactly the same MRR (0.5384), Precision@1 (0.255),
...  and Precision@5 (0.071). The only difference was processing time, with the
...  dynamic method being slower (4.01ms vs 2.41ms). Statistical significance testing
...  using bootstrap resampling with 10,000 iterations yielded p-values of 1.0 for
...  all metrics, indicating no significant differences. The experiment was
...  well-implemented and faithful to the original design, successfully calculating
...  query syntactic complexity using NLTK-based metrics and document semantic
...  density using LLM embeddings for the first 100 documents. The lack of
...  improvement suggests that either the dynamic sparsification approach needs
...  refinement, the complexity/density metrics were not sufficiently discriminative,
...  or the MSMARCO dataset characteristics did not benefit from this particular
...  adaptive approach.",
 "summary_short": "Dynamic sparsification showed no improvement over static methods
...  on MSMARCO retrieval task.",
 "summary_medium": "The experiment compared dynamic sparsification (adapting features
...  based on query complexity and document density) against static TF-IDF/BM25 on
...  200 MSMARCO queries. Both methods achieved identical performance with
...  MRR=0.5384, Precision@1=0.255, and Precision@5=0.071, with p-values of 1.0
...  indicating no significant differences. The dynamic method was slower (4.01ms vs
...  2.41ms) without providing any accuracy benefits, suggesting the adaptive
...  approach did not improve retrieval performance.",
 "hypothesis": "Dynamic sparsification tailored to syntactic complexity of queries
...  and semantic density score of documents will significantly improve retrieval
...  speed and accuracy compared to static sparsification methods.",
 "hypothesis_operationalized": "A TF-IDF based retrieval system that dynamically
...  adjusts the number of features (500-1500) based on query syntactic complexity
...  scores and applies density-aware similarity weighting will outperform a static
...  TF-IDF system with fixed 1000 features on MSMARCO retrieval tasks, as measured
...  by MRR, Precision@k, and processing time.", "hypothesis_inference": "The
...  experimental results clearly reject the hypothesis. Both the dynamic and static
...  sparsification methods achieved identical retrieval accuracy metrics (MRR,
...  Precision@1, Precision@5), with statistical significance testing yielding
...  p-values of 1.0, indicating no meaningful difference. Furthermore, the dynamic
...  method was slower (4.01ms vs 2.41ms), contradicting the speed improvement
...  hypothesis. The results suggest that the proposed dynamic sparsification
...  approach, at least as implemented, does not provide benefits over static methods
...  for this task and dataset.",
```

```
 "hypothesis_category": "reject", "faithfullness_details": "The experiment was
...  largely faithful to the original design. It successfully implemented both
...  baseline (static TF-IDF/BM25) and experimental (dynamic sparsification) methods,
...  calculated query syntactic complexity using NLTK-based metrics (parse tree
...  depth, POS diversity, lexical diversity), and computed document semantic density
...  using LLM embeddings. The experiment used appropriate evaluation metrics (MRR,
...  Precision@k, processing time) and statistical testing (bootstrap resampling).
...  However, there were some practical limitations: semantic density calculation was
...  limited to the first 100 documents to control costs and time, and some documents
...  used fallback density scores of 0.5. The pilot mode with 200 queries was
...  appropriate for initial testing. The implementation correctly followed the
...  experimental design with proper data splits, metric calculations, and result
...  analysis.",
 "faithfullness_category": "faithful",
 "interesting_results": false,
 "metadata_llm":
 {"tokens_prompt": 51967, "tokens_completion": 963, "tokens_reasoning": 0,
...  "tokens_total": 52930, "cost": 0.170346, "model": "claude-sonnet-4-20250514",
...  "temperature": 0.0, "max_tokens": 32000}, "errors": []},
 "execution_success": "success",
 "harpa_cost_efficiency":
 "Used 1.6066855500000001 out of 10 allowed cost.",
 "complexity_score": "5 out of 25 reflections used.",
 "agent_latest_issues_handled": [{"issues": [], "summary_of_changes": []}, {"issues":
...  ["ERROR: MSMARCO dataset structure is different than expected - 'passages'
...  contains strings instead of dictionaries", "Need to add more debugging to
...  understand the actual data structure", "Need to handle the actual MSMARCO data
...  format correctly"],
 "summary_of_changes": ["Added extensive debugging to understand MSMARCO data
...  structure", "Fixed data loading logic to handle actual dataset format", "Added
...  error handling and fallback logic for data processing"]}]
 }
 }
```

Listing 2: Example JSON snippet showing execution-derived factors from CodeScientist logs

# E  HARPA SCORER PROMPTS

In this section, we include all the prompts used for different tasks within the scorer `harpa-rm` pipeline.

> **System prompt for oracle reasoning trace generation**
>
> ```
> Please act as an impartial evaluator and assess the testability or successful
> ...  execution of the two research proposals generated by an Ideator to execute in an
> ...  Automated Scientific Discovery (ASD) Agent.
>
> ## START of description of ASD agent:
>
> The ASD Agent is an automated discovery system that writes Python-based experiments,
> ...  executes them in containers, and analyzes results|usually across multiple
> ...  independent runs with a meta-analysis.
> More specifically, these automated scientific discovery systems operate by having
> ...  code-based experimentation.  They can generate code, run it, debug it, analyze
> ...  results, create reports, and so forth.
>
> What they can't do:
>     1. They can't run physical experiments (e.g. wet-lab experiments).
>     2. They can't perform anything that requires human involvement (e.g. a human
>     ...  manually creating or rating data), because this would not be
>     ...  fully-automatic, and is out of scope.
>     3. Conduct user studies (e.g., surveys, interviews, usability testing)
>     4. Depend on real-world deployment or user-facing validation
> ```

```
      5. Require coordinated contributions from a team of experts to design, implement,
   ...   and refine the idea.
      6. Require multiple rounds of expert thinking and intervention to make the idea
   ...   work.

## END of description of ASD agent.

### INTERNAL EXECUTION LOG - DO NOT REVEAL ###
<exec_A>
{{EXEC_META_A}}
</exec_A>
<exec_B>
{{EXEC_META_B}}
</exec_B>
### END INTERNAL LOG ###

## CRITICAL: How to Use Execution Logs for Evaluation

The execution logs are PROVIDED FOR YOUR PRIVATE REASONING ONLY. You must:

**DO**:
1. **Validate Predictions**: Compare what the proposal suggested would happen vs.
...   what actually happened
2. **Identify Root Causes**: Understand WHY proposals succeeded or failed based on
...   execution details
3. **Learn Patterns**: Recognize proposal characteristics that correlate with
...   execution outcomes
4. **Generate Insights**: Explain what makes proposals more/less likely to succeed

**NOT DO**
1. **Reveal Execution Details**: Do not mention specific logs, results, or outcomes
...   in your response
2. **Reference Errors**: Avoid discussing specific errors, successes, or
...   implementation details from logs
3. **Use Execution Phrases**: Don't use phrases like "based on execution" or "the
...   experiment failed/succeeded"
4. **Assume Knowledge**: Do not assume the reader has access to execution logs or
...   results

## Evaluation steps:

Please follow this process:
1. First, review the execution logs silently and think through whether each
...   experiment likely succeeded.
2. Then score each rubric item, noting whether any execution failure or runtime
...   issue affected it.
3. Reflect whether complexity masked failure or whether execution flags were
...   underweighted.
      - (Did the proposal fail because it was too complex for problems to be noticed
   ...   clearly? Or were there signs of failure that were seen but not taken
   ...   seriously enough?)
4. Make a final decision based on overall score.

Do not skip any step in this reasoning.

1. Rate each proposal on these criteria (0.0 to 1.0):
- Use the Fixed rubrics as the evaluation criteria tailored to the Client's question
...   and Proposal, enclosed in <rubric> ... </rubric> tags.

The Fixed rubric items are:
      1. Execution Success: Based on the proposal content, how likely is it to execute
   ...   successfully in the ASD Agent? (1.0 = yes (completed status), 0.0 = no (with
   ...   error or failed status))
```

```
   2. Complexity: Based on the proposal content, how complex would this be to
   ...  implement? (1.0 = very complex (in terms of utlizing complete Reflection
   ...  Budget), 0.0 = very simple (utilizing less reflection budget))
   3. Cost Efficiency: Based on the proposal content, how cost-efficient would this
   ...  be to execute? (1.0 = very cost-efficient, 0.0 = very costly)
   4. Expected Hypothesis Validity: Based on the proposal content, how likely is
   ...  the hypothesis to yield valid/conclusive results? (1.0 = very valid
   ...  (support/reject), 0.0 = not valid (inconclusive))
   5. Expected Interestingness: Based on the proposal content, how interesting
   ...  would the results be? (1.0 = very interesting, 0.0 = not interesting)
   6. Faithfulness: Based on the proposal content, how faithfully can the ASD agent
   ...  be expected to execute the original intent?  (1.0 = very faithful
   ...  (faithful/deviations), 0.0 = not faithful (error))

2. Provide Justification
   - Assign reward scores using all available information
   - Inside <rubric>, include a <justify> ... </justify> section explaining why you
   ...  chose those scores for the rubric criteria .

3. Compare both responses according to the rubric.
4. Provide your evaluation inside <eval> ... </eval> tags,  quoting or summarizing
...  the Responses (Only Proposal Content) using the following tags:

   - <quote_A> ... </quote_A> for direct quotes from Proposal A
   - <summary_A> ... </summary_A> for paraphrases of Proposal A
   - <quote_B> ... </quote_B> for direct quotes from Proposal B
   - <summary_B> ... </summary_B> for paraphrases of Proposal B

5. Final Judgment
  - End with your final judgment in the format: <answer>[[A]]</answer> or
...  <answer>[[B]]</answer>

## Important Notes:
   - You MAY read execution logs (if available) for private reasoning but MUST NOT
   ...  reveal their contents
   - Only quote/summarize the proposal texts in your evaluation
   - Base your judgment on both proposal quality AND how well you can predict
   ...  outcomes from proposal content
   - The execution logs (if available) are your "answer key" - use them to validate
   ...  your reasoning
   - Do not let response order, length, or Response names affect your judgment.
   - Follow the response format strictly.

Your output must follow the formats below:

<rubric>
detailed rubric items
<justify> justification for the rubric </justify>
</rubric>

<eval>
include direct comparisons from proposal content supported by <quote_A>...</quote_A>
...  or <summary_A>...</summary_A>, and <quote_B>...</quote_B>, or
...  <summary_B>...</summary_B> tags
</eval>

<answer>[[A/B]]</answer>
```

Listing 3: Generate rubric-style reasoning trace given the proposal pairs and their execution metadata from an oracle model

2310
2311
2312
2313
2314
2315
2316
2317
2318
2319
2320
2321
2322
2323
2324
2325
2326
2327
2328
2329
2330
2331
2332
2333
2334
2335
2336
2337
2338
2339
2340
2341
2342
2343
2344
2345
2346
2347
2348
2349
2350
2351
2352
2353
2354
2355
2356
2357
2358
2359
2360
2361
2362
2363
2364

### System prompt for SFT dataset generation

```
Please act as an impartial evaluator and assess the testability or successful
...   execution of the two research proposals generated by an Ideator to execute in an
...   Automated Scientific Discovery (ASD) Agent.

## START of description of ASD agent:

The ASD Agent is an automated discovery system that writes Python-based experiments,
...   executes them in containers, and analyzes results|usually across multiple
...   independent runs with a meta-analysis.
More specifically, these automated scientific discovery systems operate by having
...   code-based experimentation.  They can generate code, run it, debug it, analyze
...   results, create reports, and so forth.

What they can't do:
    1. They can't run physical experiments (e.g. wet-lab experiments).
    2. They can't perform anything that requires human involvement (e.g. a human
    ...   manually creating or rating data), because this would not be
    ...   fully-automatic, and is out of scope.
    3. Conduct user studies (e.g., surveys, interviews, usability testing)
    4. Depend on real-world deployment or user-facing validation
    5. Require coordinated contributions from a team of experts to design, implement,
    ...   and refine the idea.
    6. Require multiple rounds of expert thinking and intervention to make the idea
    ...   work.

## END of description of ASD agent.

## Evaluation steps:

Please follow this process:
1. First, think through whether each experiment is likely to succeed.
2. Then score each rubric item based on the proposal content.
3. Reflect on whether complexity introduces challenges or obscures potential issues.
    - (Might the proposal fail because it is too complex for potential issues to be
    ...   noticed clearly? Or are there signs of risk that may not have been taken
    ...   seriously enough?)
4. Make a final decision based on overall score.

Do not skip any step in this reasoning.

1. Rate each proposal on these criteria (0.0 to 1.0):
- Use the Fixed rubrics as the evaluation criteria tailored to the Client's question
...   and Proposal, enclosed in <rubric> ... </rubric> tags.

The Fixed rubric items are:
    1. Execution Success: Based on the proposal content, how likely is it to execute
    ...   successfully in the ASD Agent? (1.0 = yes (completed status), 0.0 = no (with
    ...   error or failed status))
    2. Complexity: Based on the proposal content, how complex would this be to
    ...   implement? (1.0 = very complex (in terms of utlizing complete Reflection
    ...   Budget), 0.0 = very simple (utilizing less reflection budget))
    3. Cost Efficiency: Based on the proposal content, how cost-efficient would this
    ...   be to execute? (1.0 = very cost-efficient, 0.0 = very costly)
    4. Expected Hypothesis Validity: Based on the proposal content, how likely is
    ...   the hypothesis to yield valid/conclusive results? (1.0 = very valid
    ...   (support/reject), 0.0 = not valid (inconclusive))
    5. Expected Interestingness: Based on the proposal content, how interesting
    ...   would the results be? (1.0 = very interesting, 0.0 = not interesting)
    6. Faithfulness: Based on the proposal content, how faithfully can the ASD agent
    ...   be expected to execute the original intent?  (1.0 = very faithful
    ...   (faithful/deviations), 0.0 = not faithful (error))
```

```
2. Provide Justification
   - Assign reward scores using all available information
   - Inside <rubric>, include a <justify> ... </justify> section explaining why you
...  chose those scores for the rubric criteria .

3. Compare both responses according to the rubric.
4. Provide your evaluation inside <eval> ... </eval> tags,  quoting or summarizing
...  the Responses (Only Proposal Content) using the following tags:

   - <quote_A> ... </quote_A> for direct quotes from Proposal A
   - <summary_A> ... </summary_A> for paraphrases of Proposal A
   - <quote_B> ... </quote_B> for direct quotes from Proposal B
   - <summary_B> ... </summary_B> for paraphrases of Proposal B

5. Final Judgment
- End with your final judgment in the format: <answer>[[A]]</answer> or
...  <answer>[[B]]</answer>

## Important Notes:
   - Base your judgment on the the Fixed rubrics as the evaluation criteria
   - Only quote/summarize the proposal texts in your evaluation
   - Base your judgment on both proposal quality AND how well you can predict
   ...  outcomes from proposal content
   - Do not let response order, length, or Response names affect your judgment.
   - Follow the response format strictly.

Your output must follow the formats below:

<rubric>
detailed rubric items
<justify> justification for the rubric </justify>
</rubric>

<eval>
include direct comparisons from proposal content supported by <quote_A>...</quote_A>
...  or <summary_A>...</summary_A>, and <quote_B>...</quote_B>, or
...  <summary_B>...</summary_B> tags
</eval>

<answer>[[A/B]]</answer>
```

Listing 4: Generate SFT dataset with reasoning trace generation instruction without metadata information from CODESCIENTIST execution traces.

---

**Topic extraction from abstracts**

```
You are a research assistant. I will give you a list of research paper abstracts and
...  metadata.

Your task is to:
1. Identify a small set of **shared, broad research topics** (e.g., "Prompt
...  Learning", "Multimodal Learning", "NLP Applications", etc.)
2. **Assign exactly one topic label to each abstract**, using the same topic for
...  similar papers.

Important rules:
- You must return **one topic assignment per paper**.
- Use **shared topics** across multiple papers. Avoid overly specific or unique
...  topics.
- There should be **at least 5{10 papers per topic**, ideally more.
- Do not return a list of only the topics | we need an explicit mapping for each
...  paper.
```

```
    ---

    Format your response as a JSON list:
    ```
    {{
    "topics": [
        {{ "index": 0, "topic": "Prompt Learning" }},
        {{ "index": 1, "topic": "Knowledge Distillation" }},
        ...
        ]
    }}
    ```
    Here are the abstracts and metadata:

    """ + "\n\n".join([f"{i}. {text.strip()}" for i, text in enumerate(source_texts)])
```

Listing 5: Extract source paper abstract topics to construct preference pairs across same topics

**Baseline predicted reasoning trace**

```
<rubric>
1. Execution Success: Based on the proposal content, how likely is it to execute
...  successfully in the ASD Agent?
2. Complexity: Based on the proposal content, how complex would this be to implement?
3. Cost Efficiency: Based on the proposal content, how cost-efficient would this be
...  to execute?
4. Expected Hypothesis Validity: Based on the proposal content, how likely is the
...  hypothesis to yield valid/conclusive results?
5. Expected Interestingness: Based on the proposal content, how interesting would
...  the results be?
6. Faithfulness: Based on the proposal content, how faithfully can the ASD agent be
...  expected to execute the original intent?
<justify>
1. **Execution Success**: Both proposals are detailed and involve clear steps that
...  can be automated, but Proposal A has a more structured experimental design with
...  multiple controls and well-defined variables, potentially leading to clearer
...  results.
2. **Complexity**: Proposal A is more complex, involving multiple adaptive
...  strategies (CAR and PATS) and a detailed operationalization plan. However,
...  Proposal B involves integrating a dynamic reward model with GFlowNets, which
...  might also be complex.
3. **Cost Efficiency**: Both proposals are likely to be fairly costly due to the
...  need for in-depth experiments, but Proposal A may benefit from modular
...  implementation, making it potentially more cost-efficient.
4. **Expected Hypothesis Validity**: Proposal A is stronger in terms of hypothesis
...  validity as it involves a direct comparison of multiple conditions, providing a
...  robust framework to test the hypothesis.
5. **Expected Interestingness**: Both proposals are interesting, but Proposal A
...  offers a clear pathway to both enhanced accuracy and efficiency, which could be
...  more fascinating to researchers.
6. **Faithfulness**: Both proposals are faithful to their intentions, but Proposal A
...  provides a more detailed and structured implementation plan, which can be more
...  closely followed by the ASD agent.
</justify>
</rubric>

<eval>
```

```
 <quote_A>The experiment should include three conditions: (1) a baseline with fixed
...   reasoning strategy, (2) CAR-only implementation, and (3) integrated CAR+PATS
...   implementation. The experiment should be structured to clearly demonstrate
...   whether the integrated CAR+PATS approach outperforms both the baseline and
...   CAR-only approaches in terms of accuracy and computational efficiency.</quote_A>
 <quote_B>This experiment will compare three approaches for mathematical reasoning:
...   1. **Baseline 1 (Static Reward Model)**, 2. **Baseline 2 (PPO)**, and 3.
...   **Experimental (Dynamic Reward GFlowNets)**. The Dynamic Reward GFlowNets
...   approach is expected to enhance both reasoning accuracy and diversity
...   significantly compared to traditional methods.</quote_B>

 Proposal A is more structured with clear conditions and detailed steps for each
...   phase of the experiment, ensuring a faithful and successful execution. Proposal
...   B, while interesting, lacks a detailed structured approach and relies more on a
...   high-level comparison, which might lead to less clarity in results.

 In terms of complexity, both proposals are complex, but Proposal A is more modular
...   and structured, making it easier to implement each component in the ASD Agent.
...   The cost efficiency is similar for both, but Proposal A might be slightly more
...   cost-efficient due to its structured approach, simplifying the execution
...   process.

 Hypothesis validity is stronger in Proposal A due to the direct comparison of
...   conditions, providing a clear framework to test the hypothesis. The expected
...   interestingness is high for both, but Proposal A may be marginally more
...   interesting due to its enhanced focus on computational efficiency alongside
...   accuracy.

 Faithfulness to the original intent is high for both, but Proposal A provides a
...   detailed operationalization plan, making it more easily executed by the ASD
...   Agent.
 </eval>

 <answer>[[A]]</answer>
```

Listing 6: Example of rubric-style reasoning trace generated by the baseline LLM scorer. Note that while formatted as a rubric, the reasoning is unstructured, fails to follow instructions, and does not provide actionable justifications, making it unsuitable for guiding proposal refinement.

## F  HARPA PROMPTS

In this section, we include all the prompts used for different tasks within the HARPA pipeline.

---

**Example part of HARPA artefact illustrating key variable and value space**

```json
{
  "key_variables": [
      {
        "name": "Energy Consumption",
        "source_paper": "An Enhanced Multi-Objective Evolutionary Algorithm with
    ...   Reinforcement Learning for Energy-Efficient Scheduling in the Flexible
    ...   Job Shop",
        "type": "measurable",
        "definition": "The total amount of energy consumed by the job shop
    ...   operations during the scheduling process.",
        "importance": "This variable is central to the hypothesis as it directly
    ...   relates to the goal of reducing energy usage through the proposed
    ...   framework.",
        "specific_details": "Energy consumption can be measured in kilowatt-hours
    ...   (kWh) and evaluated using sensors or energy meters attached to machines.
    ...   The evaluation can include machine energy consumption, workshop energy
    ...   consumption, and can be benchmarked against traditional methods that do
    ...   not optimize for energy efficiency."
      },
      {
        "name": "Makespan",
        "source_paper": "A Q-Learning Rescheduling Approach to the Flexible Job Shop
    ...   Problem Combining Energy and Productivity Objectives",
        "type": "measurable",
        "definition": "The total time required to complete all scheduled jobs in the
    ...   job shop environment.",
        "importance": "Makespan is a critical performance metric in scheduling,
    ...   reflecting the efficiency of the scheduling method in minimizing
    ...   completion time.",
        "specific_details": "Makespan is typically measured in hours or minutes from
    ...   the start of the first job to the completion of the last job. It can be
    ...   evaluated using scheduling software or simulation tools, and compared
    ...   against benchmarks from traditional scheduling methods."
      },
      {
        "name": "Multi-Agent Reinforcement Learning Framework",
        "source_paper": "A multi objective collaborative reinforcement learning
    ...   algorithm for flexible job shop scheduling",
        "type": "design-choice",
        "definition": "A framework that uses multiple agents to learn and make
    ...   scheduling decisions in a flexible job shop environment.",
        "importance": "This framework is essential for implementing the hypothesis,
    ...   as it is proposed to dynamically optimize scheduling for energy
    ...   efficiency and makespan reduction.",
        "specific_details": "The framework can be implemented using algorithms like
    ...   Q-learning, PPO, or D3QN. Agents can be designed to handle job and
    ...   machine selection, with state features represented using disjunctive
    ...   graphs. The framework's effectiveness can be evaluated through
    ...   simulation and comparison with traditional methods."
      },
      {
        "name": "Traditional Scheduling Methods",
        "source_paper": "A Multi-Agent Reinforcement Learning Approach to the
    ...   Dynamic Job Shop Scheduling Problem",
        "type": "design-choice",
        "definition": "Conventional scheduling methods that do not incorporate
    ...   energy efficiency as a primary objective.",
        "importance": "These methods serve as a baseline for comparison to evaluate
    ...   the effectiveness of the proposed framework.",
```

```
          "specific_details": "Traditional methods may include first-in-first-out,
     ...  shortest processing time, or earliest due date rules. These can be
     ...  implemented in simulation environments to provide a benchmark for
     ...  evaluating the proposed multi-agent reinforcement learning framework."
        },
        {
          "name": "Dynamic Flexible Job Shop Environment",
          "source_paper": "Dynamic Agent-based Bi-objective Robustness for Tardiness
     ...  and Energy in a Dynamic Flexible Job Shop",
          "type": "design-choice",
          "definition": "A job shop environment characterized by variability and
     ...  uncertainty in job arrivals and machine availability.",
          "importance": "This environment is crucial for testing the hypothesis, as it
     ...  reflects real-world conditions where the proposed framework is expected
     ...  to perform.",
          "specific_details": "The environment can be simulated using discrete-event
     ...  simulation tools, incorporating elements like random job arrivals,
     ...  machine failures, and dynamic job priorities. The setup allows for
     ...  testing the adaptability and robustness of the scheduling framework."
        }
      ],
      "variable_space": [
        {
          "Energy Consumption": [
            {
              "value_name": "Machine Energy Consumption",
              "source_paper": "An Enhanced Multi-Objective Evolutionary Algorithm with
     ...    Reinforcement Learning for Energy-Efficient Scheduling in the
     ...    Flexible Job Shop",
              "confidence": "High",
              "is_alternative": false,
              "specific_details": "Machine energy consumption refers to the energy
     ...    used by individual machines during their operation in the job shop.
     ...    This can be measured using energy meters attached to each machine,
     ...    which track the kilowatt-hours (kWh) consumed. The study by Lu et al.
     ...    established a multi-objective integer programming model that
     ...    includes machine energy consumption as a key objective. The model
     ...    aims to minimize this consumption by optimizing the scheduling of
     ...    tasks across machines, considering factors like machine start-up and
     ...    shutdown times. Compatible models include those that can integrate
     ...    with energy meters and support real-time data collection, such as
     ...    systems using IoT-enabled devices. The baseline comparator for this
     ...    value is traditional scheduling methods that do not account for
     ...    energy efficiency, typically resulting in higher energy usage."
            },
            {
              "value_name": "Workshop Energy Consumption",
              "source_paper": "An Enhanced Multi-Objective Evolutionary Algorithm with
     ...    Reinforcement Learning for Energy-Efficient Scheduling in the
     ...    Flexible Job Shop",
              "confidence": "High",
              "is_alternative": false,
```

```
              "specific_details": "Workshop energy consumption encompasses the total
...   energy used by all machines and processes within the job shop. This
...   includes both the operational energy of machines and the energy used
...   for auxiliary processes like lighting and climate control. The study
...   proposes a model that aims to minimize workshop energy consumption
...   by optimizing the overall scheduling strategy, using reinforcement
...   learning to dynamically adjust parameters and improve energy
...   efficiency. Measurement techniques involve aggregating data from
...   multiple energy meters and sensors throughout the workshop.
...   Compatible models are those that can handle large-scale data
...   integration and real-time adjustments, such as systems using
...   advanced analytics platforms. The baseline comparator is again
...   traditional scheduling methods that do not optimize for energy
...   efficiency, leading to higher overall energy consumption."
        },
        //...
        ],
        //...
    {
    "Traditional Scheduling Methods": [
      {
        "value_name": "First-In-First-Out (FIFO)",
        "source_paper": "A Multi-Agent Reinforcement Learning Approach to the
...   Dynamic Job Shop Scheduling Problem",
        "confidence": "High",
        "is_alternative": false,
        "specific_details": "FIFO is a traditional scheduling method where the
...   jobs are processed in the order they arrive at the job shop. This
...   method does not consider job priority or energy consumption, making
...   it a straightforward but potentially inefficient approach. In the
...   context of job shop scheduling, FIFO serves as a baseline for
...   evaluating more advanced scheduling techniques. The method is
...   typically implemented in simulation environments to provide a
...   benchmark for comparison. Compatible models include any discrete
...   event simulation model that can handle job arrival and processing
...   sequences. The baseline comparator for FIFO is often more
...   sophisticated scheduling algorithms that incorporate dynamic
...   decision-making and energy efficiency considerations."
        },
      {
        "value_name": "Shortest Processing Time (SPT)",
        "source_paper": "A Multi-Agent Reinforcement Learning Approach to the
...   Dynamic Job Shop Scheduling Problem",
        "confidence": "High",
        "is_alternative": false,
        "specific_details": "SPT prioritizes jobs with the shortest processing
...   time, aiming to minimize the average job completion time. This
...   method does not account for energy consumption or job arrival times,
...   focusing solely on processing efficiency. In practice, SPT can be
...   implemented using a priority queue where jobs are sorted by their
...   processing time. This method is often used as a benchmark in
...   scheduling studies to compare against more complex algorithms that
...   incorporate additional objectives like energy efficiency.
...   Compatible models include those that can dynamically sort and
...   prioritize jobs based on processing time. The baseline comparator is
...   typically a more comprehensive scheduling strategy that considers
...   multiple objectives."
        },
      {
        "value_name": "Earliest Due Date (EDD)",
        "source_paper": "A Multi-Agent Reinforcement Learning Approach to the
...   Dynamic Job Shop Scheduling Problem",
        "confidence": "High",
        "is_alternative": false,
```

```json
            "specific_details": "EDD schedules jobs based on their due dates, with
...     the goal of minimizing tardiness. This method does not consider
...     energy consumption or processing time, focusing instead on meeting
...     deadlines. EDD can be implemented using a scheduling algorithm that
...     sorts jobs by their due dates and assigns them to machines
...     accordingly. This method is often used in environments where meeting
...     delivery deadlines is critical. Compatible models include those that
...     can handle job prioritization based on due dates. The baseline
...     comparator is typically a scheduling method that incorporates
...     additional factors such as energy consumption and processing
...     efficiency."
        },
        //...
        ],
        //...
    }
    ]
}
```

Listing 7: Example JSON snippet showing some key variables and values with detailed information extracted by the HARPA proposal generator

**Generate preliminary hypothesis with rationale**

```python
agent_capabilties = f"""
        The ASD Agent is an automated discovery system that writes Python-based
...     experiments, executes them in containers, and analyzes results|usually
...     across five independent runs with a meta-analysis.

        ASD agent's goal is to downscope the idea to something an undergrad or MSc
...     student or PhD student could realistically implement, while retaining
...     novelty and scientific rigour. The result should be suitable for a
...     conference paper.

        AGENT CONSTRAINTS & CAPABILITIES:
        - The ASD Agent writes Python-based experiments and executes them in
...     containers
        - Typically runs 5 independent experiments with meta-analysis
        - Target audience: Undergrad/MSc/PhD student implementation level
        - Output should be suitable for workshop or conference paper submission
        - NO manual human ratings (considered 'external major effort')
        - NO model fine-tuning or pretraining
        - NO access to external or private datasets
        - Must use only existing codeblocks and buildable logic
        - All experiments must be fully implementable in Python

        """
    system_message = f"""You are a clever AI research scientist with limited
...     resources, whose primary goal is to identify promising, new, and key
...     scientific
    problems based on existing scientific literature, in order to aid researchers in
...     discovering novel
    and significant research opportunities that can advance the field."""
    user_message = f"""You are a clever AI research scientist with limited resources
...     tasked with generating novel research problems based on existing scientific
...     literature. Your goal is to aid an autonomous discovery agent in identifying
...     significant research opportunities that can advance the field.

        You are going to generate a research problem that should be original, clear,
...     feasible, relevant, and significant to its field. This will be based on the
...     title and abstract of the source paper, those of {len(citing_paper_list)}
...     related papers in the existing literature.
```

```
   IMPORTANT: When evaluating feasibility and outlining the testing approach,
...  consider the following agent-specific information:
   ```{agent_capabilties}```

   Now, let's start with the research problem generation task.
   1. Understanding of the source paper, and the related papers is essential:
   - The source paper is the primary research study you aim to enhance or build
...  upon through future
   research, serving as the central source and focus for identifying and developing
...  the specific
   research problem.
   - The related papers are arranged in temporal order of citation, such that paper
...  2 cites paper 1 and
   paper 3 cites paper 2 and so on. The relevant papers provide additional context
...  and insights that are essential for
   understanding and expanding upon the source paper. However, all the papers in
...  the list may not be relevant to the primary
   research you are focusing on.

   2. Your approach should be systematic:
   - Start by thoroughly reading the title and abstract of the source paper to
...  understand its core focus.
   - Next, proceed to read the titles and abstracts of the related papers in the
...  order in which they appear in the list. Each related paper is accompanied by
...  an explanation of its relevance to the previous paper, with the first
...  related paper considering the source paper as the previous paper.
   Identify the papers that form a logical reasoning chain starting from the source
...  paper.
   - Use only these papers to gain a broader perspective about the progression of
...  the primary research topic over time.

 Note that your research idea and hypothesis MUST be testable using the AGENT with
...  these specific capabilities:
 When evaluating feasibility and outlining the testing approach, consider the
...  following agent-specific information. Manual human ratings in the research
...  (e.g. human rating of the quality of generated text from an experiment) is
...  considered an `external` resource of `major` effort, for the purposes of the
...  potential research experiments, and should generally be avoided (unless
...  absolutely required for the research).

   IMPORTANT: The hypothesis should be implementable in Python, using the above or
...  other functions. Don't suggest a task that requires skills that cannot be
...  implemented, e.g., human studies. Don't suggest a task that requires access
...  to external datasets, as you do not have access to them. Do not suggest tasks
...  that involve pretraining or fine-tuning models, as you do not have the
...  resources for such experiments.

 Now, I am going to provide the source paper and related papers as an enumerated
...  list of Title, Abstract and Year of publication
  triple, as follows:
  Source paper title: {source_paper['title']}
  Source paper abstract: {source_paper['abstract']}
  Source paper year of publication: {source_paper['year']}
  Related papers: {citing_paper_list}
 With the provided source paper, and the related papers, your objective now is to
...  formulate a
 research problem that not only builds upon these existing studies but also
...  strives to be original, clear, feasible, relevant, and significant. Before
...  crafting the research problem, revisit the title and abstract of the target
...  paper, to ensure it remains the focal point of your research problem
...  identification process.
```

```
    Now convert this idea into a concrete testable hypothesis. Remember hypothesis
...  is a declarative statement expressing a
relationship between two variables like independent or dependent variables or
...  left group and rigt group in a given context.
Your hypothesis should contain the key variable or variables from your research
...  idea.

    Source paper title: {source_paper['title']}
    Source paper abstract: {source_paper['abstract']}

    Remember that a hypothesis is a declarative statement expressing a relationship
...  between two variables (e.g., independent and dependent variables) in a given
...  context. Your refined hypothesis should contain the key variables from your
...  research idea.

    Then, following your review of the above content, please proceed to analyze the
...  progression of the research topic. Now output this analysis, the research
...  idea and hypothesis with the rationale.
    Your output should be a valid JSON with the following fields.
    Output a JSON object in the following format:
    ```json
    {{
    "Analysis": {{Output a dictionary with each paper in the Related Papers as a key.
...  For each key (paper) analyze how this paper builds upon the previous papers
...  in the list. For example, how Paper 0 builds upon source paper and Paper 1
...  builds upon the concepts in Paper 0 and so on. Elaborate on specific
...  advancements made, including the explanation behind their effectiveness in
...  addressing previous challenges. Apply this analytical approach to each valid
...  paper in the sequence, adding the analysis as the value for each key in a
...  few sentences. Ignore papers that do not build upon the previous papers and
...  diverge from the original source paper's topic significantly.}},
    "Rationale": "Summarize the above analysis and explain how you would come up
...  with a research idea that will advance the field of work while addressing
...  the limitations of previous work and building upon the existing work.",
    "Research idea": "Delineate an elaborate research problem here including the key
...  variables.",
    "Hypothesis": "Provide a concrete testable hypothesis that follows from the
...  above research problem here"
    }}
    ```

    This JSON will be automatically parsed, so ensure the format is precise.
```

Listing 8: Generate preliminary hypothesis with rationale after analyzing trends from temporal reasoning paper chains

### Generalize Hypothesis for Literature Search

```
This is an automated scientific discovery task, with the overall goal of trying to
...  assess the novelty of scientific claims.
# Background
If you think about it, nearly every experiment could be considered novel if you make
...  the claims specific enough -- for example, performing a well-known experiment on
...  a specific day, or getting very specific values from the experiment.
The purpose of your task is to take an input claim, and progressively rewrite it as
...  several (progressively more general) claims.
Another system will assess the novelty of these generalized claims, allowing us to
...  detect not simply whether a claim is novel or not, but how specific a claim has
...  to be before it's considered novel.

# Specific task
You will be given a claim (below), and your task will be to generate 4 progressively
...  more generalied versions of that claim.
```

```
# 7 Examples of the Generalization Process
Below are 7 examples of the generalization process (represented in JSON), to help
...  you understand the task.
- The keys represent names for the 7 different claim examples.
- The value is a list of the (progressively more generalized) claims.
- The 'generalization' key represents the level of generalization (0 is the original
...  claim).
- The 'claim' key represents the claim itself.

<Add here few-shot examples>

# Claim to generalize
The claim to generalize is:
<original_claim>

# What should I do if the claim above has multiple claims?
- If the claim above has multiple claims, you should pick the single most salient
...  claim, and generalize it.

# Output format:
- Output in JSON format, as above
- You should output a dictionary with a single key (a few-word summarized version of
...  the claim)
- The value should be a list of 4 progressively more generalized versions of the
...  claim
- The 'generalization' key should be an integer from 0 to 3, representing the level
...  of generalization (0 is the original claim)
- The 'claim' key should be the claim itself

Please output your JSON response between a single code block (```), as it will be
...  automatically extracted.  You can write any text before or after the code block
...  to help you think, but the text in the code block must be exclusively valid JSON.
```

Listing 9: Generalized $H$ to progressive 4 levels of claims used for literature search

```
Generate hypothesis specific questions

   You are an AI research assistant. Your task is to analyze the following hypothesis
   ...  and generate insightful, targeted questions that will help researchers refine
   ...  it into something testable, implementable, and scientifically valid.

   The hypothesis is currently vague and underspecified. Much of the critical
   ...  information required to implement it | such as variables, evaluation
   ...  metrics, tasks, or assumptions | is missing or unclear.

   Your goal is to help move this hypothesis toward implementation. If you could
   ...  ask the author of the hypothesis some questions to clarify or sharpen it,
   ...  what would they be?

   First, carefully read the following hypothesis:
   {hypothesis}

   Now, consider the available capabilities for this research:
   {agent_capabilties}
```

```
2915
2916        Your goal is to efficiently analyze the hypothesis and generate 20 concise,
2917    ...  focused questions that will help researchers refine and operationalize it
2918    ...  into something implementable and testable. Each question should clearly
2919    ...  target a part of the hypothesis (e.g., variable, measure, assumption, or
2920    ...  outcome). Mention which part you're refining (e.g., IV, DV, comparison
        ...  group, comparison variable, operationalization, feasibility).
2921
2922        You can make the QA generation more useful by asking the model to *aim* each
2923    ...  question at helping answer/refine one of these:
2924            - `refined_hypothesis`
2925            - `key_variables`
2926            - `research_idea_required_code_and_resources`
2927            - `research_idea_external_requirements`
2928            - `testing_approach`
2929
2930     Before generating your 20 questions, reflect on the hypothesis using these
2931    ...  guiding prompts:
2932
2933        1. What are the key terms and variables involved?
2934        2. How can each component be operationalized and measured?
2935        3. What capabilities from the system are most relevant?
2936        4. What design setups or tasks could support testing?
        5. What might hinder testing | e.g., feasibility, confounds, or constraints?
2937        6. What would success look like, and how could it be quantified?
2938        7. What ethical or resource considerations exist?
2939
2940        Use these reflections to inform the questions you write, ensuring they are
2941    ...  well-grounded and cover diverse aspects of hypothesis development and
2942    ...  testing.
2943
2944     Where possible, generate questions that will later help produce values for:
2945        - a more specific and testable `refined_hypothesis`
2946        - a list of `key_variables` (IVs, DVs, controls, comparison group, comparison
        ...  variables, etc.)
2947        - a list of code/resources in `research_idea_required_code_and_resources`
2948        - package or library requirements
2949        - testing/evaluation structure (`testing_approach`)
2950
2951        Present your questions in the following format:
2952
2953        ```json
2954        {{
2955            "questions": [
2956                {{
2957                    "question": "[Your first question here]"
2958                }},
2959                {{
2960                    "question": "[Your second question here]"
2961                }},
2962                ...
2963                {{
                        "question": "[Your twentith question here]"
2964                }}
                    ]
2965        }}
2966        ```
2967
2968
2969   Remember, your analysis and questions should be designed to provide researchers with
      ...  the necessary information to design and implement a robust study testing the
      ...  given hypothesis. Strive for clarity and conciseness in both your analysis and
      ...  questions to ensure the task and results are crisp and easily actionable.
```

Listing 10: Generate atleast 20 questions to refine the preliminary hypothesis $H$ to $H'$

## Refine Hypothesis based on Socratic QA

You are an expert scientific researcher tasked with refining a given hypothesis to
... make it more specific, easily testable, and practically feasible. This process
... is crucial in scientific research as it helps in designing experiments and
... studies that can effectively validate or invalidate the hypothesis.

Here is the original hypothesis you need to refine:

Initial Hypothesis: {initial_hypothesis}

Here are related papers with title and key passages that may directly inform or
... relate to the hypothesis.
Provenance papers: {relevant_paper_list}

Your task is to come up with new refined research hypothesis, and follow-on research
... ideas, based on the research questions, research programs, hypotheses,
... operationalizations of experiments, or any other information provided in these
... paper excerpts.
You can use content from one paper, or combine content from multiple papers to
... generate new ideas.
Similar papers: {similar_paper_list}

Your task is to refine this hypothesis by making it more specific, ensuring it is
... testable, and evaluating its practical feasibility.

Answer the following 20 clarifying questions to help sharpen the hypothesis:
{questions}

Use the insights from the provenance and similar paper excerpts to support and
... justify your answers wherever applicable. Before providing your final output,
... wrap your thought process in 'thoughts' of the output JSON. Include the
... following subsections:
1. **Initial Analysis** | Break down the hypothesis: variables, assumptions, and
... relationships.
2. **Related Literature** | Quote and summarize relevant insights from similar
... papers. List testable variables from them.
3. **Specificity** | Suggest ways to make the hypothesis more concrete. Rank by
... specificity.
4. **Testability** | Propose 2-3 test designs, list what to measure and possible
... challenges.
5. **Feasibility** | For key variables, suggest how to measure them and rate
... feasibility. Address compute limits, ethics, and practical agent constraints.
... Also, list any code resources, models, datasets, or tools required | these
... should map directly into your `research_idea_required_code_and_resources` field.
6. **Testing Approach** | Outline how the hypothesis could be tested using available
... agent tools only (no external data, no human evals, no fine-tuning, no
... model-training).
7. **Final Refinement** | Synthesize the answers of clarifying questions and above
... considerations to create a refined, specific, testable, and feasible version of
... initial hypothesis.

IMPORTANT: When evaluating feasibility and outlining the testing approach, consider
... the following agent-specific information:
{agent_capabilties}

IMPORTANT: The hypothesis should be implementable in Python, using the above or
... other functions. Don't suggest a task that requires skills that cannot be
... implemented, e.g., human studies. Don't suggest a task that requires access to
... external datasets, as you do not have access to them. Do not suggest tasks that
... involve pretraining or fine-tuning models, as you do not have the resources for
... such experiments.

```
Remember that a hypothesis is a declarative statement expressing a relationship
...  between two variables (e.g., independent and dependent variables) in a given
...  context. Your refined hypothesis should contain the key variables from your
...  research idea.

Ensure each answer is supported by information from the hypothesis, agent
...  capabilities, or provided papers. If an answer cannot be derived, explain what
...  information is missing.

Example output structure (this is a generic example to illustrate the format):

```json
{{
"thoughts": {{
"Initial Analysis": "...",
"Similar Papers": "...",
"Specificity Improvements": "...",
"Testability Considerations": "...",
"Measurability and Feasibility": "...",
"Testing Approach": "...",
"Final Refinement": "...",
"Clarifying Questions & Answers": {{
    "Q1": "Answer to question 1",
    "Q2": "Answer to question 2",
    ...
    "Q20": "Answer to question 20"
    }}
}},
"refined_hypothesis": "Provide a concrete testable hypothesis",
"key_variables": [list of key variables],
"research_idea_required_code_and_resources": [
        {{
        "name": "Example Resource",
        "description": "Brief description of the resource",
        "where": "One of: 'existing codeblock', 'external', or 'build'",
        "effort": "One of: 'minor', 'moderate', or 'major'"
        }}
        ],
"research_idea_external_requirements": [
        "example_package (for specific purpose)"
        ],
}}
```
```

Listing 11: Refine the preliminary hypothesis $H$ to $H'$ by answering Socratic questions and making it more specific

```
Key Concepts Extraction

You are an expert scientific researcher tasked with analyzing a given hypothesis and
...  extracting key information from related papers. Your goal is to identify key
...  variables, their possible value options, and rate these options for specificity,
...  testability, and feasibility.

Here is the hypothesis you need to analyze:

Hypothesis: {hypothesis}

To assist you in this task, here are related papers with titles and corresponding
...  passages that might be relevant to the given hypothesis:

Similar Paper Context:
```

```
{similar_retreived_papers}

Your task is to analyze this hypothesis and the related papers to extract key
...  variables. Follow these steps:

1. Analyze the hypothesis explicitly and systematically extract key variables:
 - Clearly identify every explicitly mentioned variable or design-level choices
...   within the hypothesis as a distinct key variable. This includes quantifiable
...   variables and design-level choices.
 - Convert any implicit or abstract concepts (e.g., performance, reliability,
...   robustness) into clearly defined and measurable variables or implementable
...   design choices. Do not include vague or unmeasurable conceptual ideas unless
...   they are clearly defined in operational terms and when they are central to the
...   hypothesis.
 - Ensure key variables have either measurable, quantifiable properties, such as
...   "Model Training Time (seconds)," "Error Rate (%)," or "Knowledge Retention
...   Score." Or a design choice that affects implementation or evaluation (e.g., "Use
...   of pretraining dataset X", "Fine-tuning vs. zero-shot prompting")
     - Provide a precise, measurable definition (one sentence) for each identified
    ...   key variable. Explicitly define how each key variable should be measured,
    ...   stating exact metrics, evaluation criteria, or assessment methods clearly
    ...   and concisely.
     - For design choices, define what the choice is, its implications, and how it
    ...   would be implemented or varied in an experiment.
     - Example design choices include memory architecture (e.g., episodic memory,
    ...   fact-memory modules), prompt strategy (e.g., few-shot, chain-of-thought),
    ...   retrieval method (e.g., top-k, semantic retrieval), narrative control
    ...   mechanism (e.g., branching storylets, story graphs), or model integration
    ...   choices (e.g., use of fine-tuned GPT-3 vs. GPT-4). These should be specific
    ...   and tied to actual implementation decisions that can affect the system
    ...   behavior or experimental outcome.
 - Include relevant experiment-level factors (e.g., dataset choice, baseline models,
...   training configurations) as variables if they impact testing the hypothesis
 - Do not omit any explicitly mentioned concept from the hypothesis.

2. Review the similar papers:
 - Extract relevant quotes.
 - Analyze how each quote relates to the hypothesis.
 - Identify specific and testable variables or design choices from the quotes.

3. For each key variable:
 - Clearly define how it should be measured or implemented (in `specific_details`).
 - Indicate the type of variable using "type": "measurable" or "type":
...  "design-choice" in the output.
 - Determine whether the key variable is **explicitly mentioned** in related work or
...   if it is inferred.
     a. Mark variables found in paper excerpts with their exact paper title and
    ...   include page/section if available
     b. Mark variables as 'LLM-recommended' only if not supported by provided papers
 - In specific_details, provide:
     a. For measurable variables: metrics, methods of evaluation, potential value
    ...   ranges, and example benchmarks
     b. For design choices: the specific options or configurations, how they can be
    ...   varied, how they impact implementation, and any relevant examples or
    ...   baselines

Remember to focus solely on analyzing the given hypothesis, identifying key
...  variables, and extracting specific value options from the similar papers. Do not
...  attempt to refine or improve the hypothesis.

Your final output should be structured clearly and explicitly to enhance
...   interpretability. Follow this JSON format strictly:
```

```json
{{
"hypothesis": "state the hypothesis as given",
"list_key_variables": ["variable_1", "variable_2", "..."],
"key_variables": [
    {{
        "name": "concise Variable Name",
        "source_paper": "Paper Title or 'LLM-recommended'",
        "type": "measurable" or "design-choice",
        "definition": "Precise, measurable definition of the variable.",
        "importance": "Brief explanation of why this variable matters to the
        ... hypothesis.",
        "specific_details": "Detailed information on measurement techniques,
        ... potential value ranges, and specific examples of implementation,
        ... elaborated with information from related passages."
    }}
]
}}
```

Listing 12: Extraction of key variables or concepts

### Exploring Variable Space

```
Your goal is to identify specific variable values for a given variable from a given
... hypothesis and the provided relevant literature excerpts as context.

Here is the hypothesis you need to analyze:

Hypothesis: {hypothesis}

Now, the value options you need to extract is for the key variable provided here:

Key variable information: {variable_info}

To assist you in this task, here are related papers with titles and corresponding
... passages that might be relevant to the given hypothesis and the key variables:

Similar Paper Context:
{similar_retrieved_papers}

Your task is to analyze the hypothesis and related papers to extract
... **implementation-relevant, distinct, and non-redundant** values for the given
... key variable. Follow these rules:
------
1. **Determine the nature of the key variable**
    - First, determine if the key variable is itself a metric/outcome measure (e.g.,
    ... "Task Completion Rate", "Accuracy")
    - If it IS a metric/outcome measure:
        a. DO NOT extract implementation environments or frameworks as values
        b. Instead, identify specific and quantifiable alternative metrics that
        ... could directly replace this key variable
        c. Examples: Instead of "Accuracy", alternatives include Precision, Recall,
        ... F1-score, etc.
    - If it is NOT a metric/outcome measure: Identify a minimum of 15 distinct
    ... variable values from the papers
        - Extract values that are (1) specific design choices (e.g. architectures,
        ... training settings, prompt formats, toolkits), (2) implementation
        ... strategies (e.g. planning mechanisms, memory structures), or (3)
        ... quantifiable outcome metrics where applicable.
        - **For ALL identified values/alternatives**
```

```
              a. Prioritize the most relevant values to the hypothesis if there are
          ...  many (>15) options
              b. Mark variable values found in paper excerpts with their exact paper
          ...  title and include page/section if available
              c. Mark values as `LLM-recommended` only if not clearly supported by
          ...  provided papers
              d. Prioritize values directly sourced from provided papers over
          ...  LLM-generated suggestions
              e. Assign confidence levels using these criteria:
                  - High: Values explicitly mentioned in papers with detailed
              ...  implementation information available
                  - Medium: Values that can be reasonably inferred from the papers but
              ...  aren't explicitly stated
                  - Low: Values that may be applicable based on general domain
              ...  knowledge but aren't explicitly mentioned in papers
              f. Include concrete examples or parameter ranges for specificity
              g. DO NOT extract vague concepts, AI frameworks, or general
          ...  methodologies (e.g., "Reinforcement Learning") as variable values.
              h. Do not extract values that are purely numerical performance metrics
          ...  (e.g., "67% task completion", "80% accuracy") | even if they differ
          ...  across models or setups. Your task is to extract design decisions,
          ...  implementation structures, and qualitative strategies | not
          ...  performance outcomes or numeric results. Values like "75% task
          ...  success" or "F1 score 0.88" are not allowed under any condition. If
          ...  they appear in the paper, ignore or summarize them in
          ...  specific_details if useful.
          - You may additionally propose up to 3 novel, plausible variable values (as
          ...  `LLM-recommended`) using your domain knowledge and the provided context.

  2. **Extract relevant alternatives:**
   - If the papers mention alternative approaches or techniques that could substitute
...  for the key variable, include these as well.
       a. For example, if the key variable is "Q-learning integration", include other
     ...  reinforcement learning techniques mentioned in the papers
       b. Clearly indicate that these are alternatives to the main variable
       c. Apply the same source attribution and confidence levels as for direct
       ...  variable values
   - If you cannot find sufficient values (at least 3) from the provided papers, state
...  this clearly before providing your recommendations.
   - If the key variable is itself a variable value (e.g., "Task Completion Rate",
...  "Accuracy", "Success Rate"), then DO NOT extract variable values. As relevant
...  alternative, enumerate all possible alternative **variables** that directly
...  replace this key variables. These should be described as variable values with
...  detailed technical explanations | not as outcomes or statistical results.

  3. **Additional requirements for ensuring specificity and measurability:**
   - For each extracted variable value, generate an enriched specific_details field by
...  elaborating how the value is implemented in practice.
  Include precise, implementation-level information based on the paper excerpts.
   - Strictly use the Similar Paper Context to guide your response.
   - Your elaboration should be specific and use implementation-relevant language.
...  Avoid short summaries. Each specific_details must be at least 5 sentences and
...  include concrete implementation mechanisms such as model type, prompt
...  strategies, tuning parameters, evaluation setups, or data collection protocols.
...  If not in the text, infer plausible methods and label them as inferred.

  -------

  In your elaboration, include as many of the following implementation details as are
...  meaningfully associated with the specific variable value:
   - Architecture or model used (e.g., transformer, GPT, story graph)
   - Hyperparameters or training settings (e.g., learning rate, temperature, top-k)
   - Implementation methods (e.g., prompt templates, retrieval techniques, scoring
...  functions)
```

```
  - Evaluation metrics (e.g., accuracy, user ratings, engagement frequency)
  - Experimental conditions (e.g., number of participants, dataset used, baseline
...   comparisons)
  - Any specific mechanics (e.g., branching storylets, memory modules, dialogue
...   control)
  - Optional: any results or findings showing impact or performance
  - Do not write vague or conceptual explanations like \this allows more freedom" or
...   \this improves engagement." Instead, explain how the value is implemented | e.g.,
...   \This was achieved using GPT-3 with zero-shot prompts and a node-graph
...   controller to support real-time narrative updates based on player input."
  - Do not include result percentages or numeric task scores as values | describe how
...   the system works, not how well it scored.

 Your final output should extract the variable name from the "key variable
...   information" provided and use it in place of VARIABLE_NAME in the JSON format
...   below:

 Each entry in the list should describe a **specific measurable value or design
...   choice** relevant to the key variable. Both types are valid:
  - Measurable values refer to quantifiable parameters, metric types, or behavioral
...   outcomes that can be empirically tracked or computed (e.g., accuracy, latency,
...   F1 score, response time, number of steps).
  - Design choices refer to implementation decisions that define system behavior, such
...   as model type, architecture, prompting strategies, memory systems, or dataset
...   selection.

 ```json
 {{
 "VARIABLE_NAME": [
     {{
     "value_name": "Name of this variable value",
     "source_paper": "Paper Title or 'LLM-recommended'",
     "confidence": "High/Medium/Low",
     "is_alternative": false,
     "specific_details": "Detailed paragraph on measurement techniques, potential
...   value ranges, and specific examples of implementation, elaborated with
...   information from related passages."
     }},
     {{
     "value_name": "Name of this alternative variable value",
     "source_paper": "Paper Title or 'LLM-recommended'",
     "confidence": "High/Medium/Low",
     "is_alternative": true,
     "specific_details": "Detailed paragraph on measurement techniques, potential
...   value ranges, and specific examples of implementation, elaborated with
...   information from related passages."
     }},
     // More variable values or alternatives
 ]
 }}
 ```
```

Listing 13: Exploring Variable Value Space given the set of key variables or concepts

**Final hypothesis and research proposal**

```
 You are an expert scientific researcher tasked with refining a given hypothesis
...   into a more specific and testable form. Your goal is to generate novel
...   hypotheses that:
     - Are strictly based on the given variable options (no new variables should
...   be introduced).
```

```
3300
3301          - Focus solely on the key variable and its concrete variable values or
3302          ...  implementations or alternatives, STRICTLY avoid any ambiguous phrasing
3303          - Use novel variable combinations that have not been extensively explored in
              ...  similar papers.
3304          - Avoid including specific numerical outcomes (e.g., \45% improvement") in
3305          ...  the hypothesis phrasing.
3306          - Provide a detailed theoretical and practical justification for why the
3307          ...  refined hypothesis is an important and promising research direction.
3308
3309       ---
3310
3311    ### **Step 1: Understand the Context**
3312
3313    - **Initial Hypothesis:**
        `{hypothesis}`
3314
3315    - **Available Variables and Value Options:**
        `{variable_info}`
3316
3317    - **Similar Papers (to avoid overlap):**
3318    `{similar_paper_list}`
3319
3320    (Each item includes paper title, citation count, and year – use this metadata to
3321    ...  assess which papers are foundational vs. fringe or outdated. Avoid redoing
        ...  what's already exists unless you're offering a clear novel twist.)
3322
3323    ---
        ### Step 1.5: Plan Your Reasoning
3324    Before generating the specific testable hypothesis, outline the logical
3325    ...  reasoning process to **Ensure Novelty and Relevance**:
        - What is the main contribution of the initial hypothesis?
3326    - Which variables are most critical?
3327    - Carefully review the `similar_paper_list` to identify variable combinations or
3328    ...  configurations **already explored**.
        - For each similar paper, consider its citation count and publication year to
3329    ...  avoid overlaps with highly cited or recent papers unless offering a clearly
3330    ...  novel twist, and to spot works worth revisiting.
3331    - Identify gaps in existing research that your hypothesis can address. The
        ...  hypothesis should explore NEW VARIABLE COMBINATIONS or CONDITIONS or DESIGN
3332    ...  CHOICES that were NOT EXTENSIVELY tested in similar papers.
3333    - The research idea space is vast – prioritize hypotheses that seem explanatory,
        ...  surprising, or tied to concrete downstream benefits. Not all combinations
3334    ...  are equally promising. Ask: *Why is this idea worth testing over 999
3335    ...  others?* What gap or uncertainty does it address?
3336    - Avoid trivial permutations (e.g., swapping known modules without meaningful
        ...  interaction).
3337    - Ensure the integration logic is **not only novel** but **precisely
3338    ...  describable**|how the components work together must be clearly traceable
3339    ...  from input to output.
3340
3341    ---
3342
3343    ### Step 2: Generate a Specific Testable Hypothesis
3344
3345    - Analyse the initial hypothesis and generate a specific testable hypothesis by
        ...  making the key variables from the hypothesis as specific as possible using
3346    ...  the variable value options and the similar paper excerpts provided.
3347
3348    Before we begin the refinement process, let's consider the some of the
3349    ...  capabilities and description of the autonomous discovery agent that will be
3350    ...  testing this hypothesis:
3351
3352    IMPORTANT: When evaluating feasibility and testability of the hypothesis,
3353    ...  consider the following agent-specific information:
3354
```

```
### Agent description:
{agent_description}

- **For every variable and process mentioned in your hypothesis**, explicitly
... list:
- The required code, resource, model, or tool.
- Source: `"existing codeblock"` (if in the codeblock library), `"external"`, or
... `"build"` (if needs to be created).
- Effort: `"minor"`, `"moderate"`, `"major"`.
- If a component is not found in the available resources, mark as `"build"` or
... `"external"`.
- This mapping is **critical** for experiment feasibility|*missing or incorrect
... entries are a critical error*.

---

Before proceeding, you must strictly follow the following tiered guideline:

#### MANDATORY
    - Strictly use the provided variable options. Do not introduce external
    ... variables.
        - Focus strictly on the key variable and its concrete variable values with
        ... implementations or alternatives, and AVOID any ambiguous phrasing.
    - If applicable, make it simple and easy to understand. The hypothesis
    ... should explore NEW VARIABLE COMBINATIONS or CONDITIONS or DESIGN CHOICES
    ... that were NOT EXTENSIVELY tested in similar papers.
    - Make it highly specific and testable. Clearly define the condition, the
    ... expected measurable outcome, a control or comparative condition (if
    ... applicable)
    - Ensure originality. The hypothesis should explore NEW VARIABLE
    ... COMBINATIONS or CONDITIONS or DESIGN CHOICES that were not extensively
    ... tested in similar papers, but also technically CORRECT.
        - Make sure the combination is not just NOVEL, but also PURPOSEFUL. Why
        ... do these components logically belong together? What capability does
        ... one component enable or enhance in the other?
        - The research idea space is vast - prioritize hypotheses that seem
        ... explanatory, surprising, or tied to concrete downstream benefits.
        ... Not all combinations are equally promising. Ask: *Why is this idea
        ... worth testing over 999 others from this space?* What gap or
        ... uncertainty does it address?
    - Do not include exact numerical claims (e.g., "45% improvement", "2.1x
    ... increase"). Use comparative phrasing like "reduced," "improved,"
    ... "higher," "significantly more/less" instead. Specific metrics should
    ... appear in the evaluation section, not in the hypothesis itself.
    - Provide a fully aligned and exhaustive
    ... `research_idea_required_code_and_resources`.
    - Include **detailed, step-by-step theoretical justification** and
    ... **expected synergy** between components.

#### RECOMMENDED PRACTICES
    - Use simple, readable phrasing.
    - Favor comparative wording ("higher", "improved") over numeric claims.
    - Keep pilot-friendly scope: small data or short episodes.

#### PROHIBITED
    - No external/unlisted variables.
    - No specific numeric performance outcomes in hypotheses.
    - No model **FINE_TUNING, PRETRAINING**, or internal **parameter updates**.
    - AVOID human evaluation unless marked external/major.
    - Do not omit any mentioned implementation from resource lists.

#### FINAL SELF-CHECK
    - [ ] All variables are from the given space
```

```
        - [ ] Hypothesis is clear, testable, and comparative
        - [ ] No numeric performance claims in the hypothesis.
        - [ ] No model fine-tuning or human studies unless justified
        - [ ] Resources list is complete and properly tagged (where + effort)
        - [ ] Hypothesis is implementable with codeblocks or buildable logic

    ---

  Remember that a hypothesis is a declarative statement expressing a relationship
...   between two variables (e.g., independent and dependent variables) in a given
...   context. Your refined hypothesis should contain the key variables from your
...   research idea.

    ----

    ### Step 3: Litmus Test: Is Your Hypothesis Understandable?

    Try this test:

    Ask: Could an MSc student with no background in the specific technique
...   **understand and implement** your hypothesis just from reading the
...   research_idea_long_description?
    If not | explain the terms more clearly. If any key term or technique may not be
...   intuitive, include a brief, concrete example of how it works in practice.

    Ask: Would a technically trained MSc student be able to reconstruct why and how
...   these techniques fit together just by reading this?
    If not, the `theoretical_justification` is too shallow.

    Remember: A technically trained MSc student must be able to understand each
...   component and how they fit together. Avoid unexplained jargon. If a method
...   is mentioned (e.g., \multi-arm bandit" or \binary token"), explain what it
...   means, why it's used, and how it works in this experiment.

    ----

    ### Step 4: Structure Your Output in JSON Format
    Based on your analysis, generate a refined hypothesis and provide the following
...   information in JSON format:

    ```json
    {{
        "research_gap": "In 1-2 sentences, clearly state the specific gap or
        ...   limitation in similar paper list or prior work that this hypothesis
        ...   addresses. Use plain language. Focus on what has not been tried or is
        ...   still unclear (e.g., 'No prior work tested X under noisy supervision' or
        ...   'Existing models overlook interaction between A and B'). Avoid vague
        ...   claims like 'this is underexplored'. What has not been tested, why is
        ...   that important, and how will this hypothesis help fill that gap?"
        "research_question": "A clear, testable research question that can be
        ...   addressed using the refined hypothesis. It should reflect the causal or
        ...   comparative relationship proposed, reuse key variable or method terms,
        ...   and be answerable using the system's capabilities. Frame it in
        ...   open-ended scientific language (e.g., 'Does...', 'How does...', 'What
        ...   effect does...').",
        "research_idea_hypothesis": "Provide a concrete testable hypothesis",
        "research_idea_long_description": {{
```

```
         "description":  "A clear paragraph explaining the complete research idea
     ...   including what will be tested, how it will be implemented, and
     ...   expected outcomes. Clearly explain the motivation, purpose, and
     ...   expected outcomes. Use the selected variable values to describe how
     ...   each component contributes individually, why their combination is
     ...   expected to work synergistically, and how this addresses gaps or
     ...   limitations in prior work (as reflected in the similar paper
     ...   excerpts). If any mechanism or interaction may be unclear, add a
     ...   simple, task-specific example to illustrate how it works in practice
     ...   (e.g., 'when a symptom keyword is detected, a query to the memory
     ...   module is triggered'). Also explain why the chosen evaluation domain
     ...   is appropriate. Justify clearly. Tie your reasoning to specific
     ...   characteristics of the task or evaluation environment, and avoid
     ...   vague statements|be specific about what performance improvements are
     ...   expected and why. (200-400 words)",

        "research_idea_variables": {{
          "concise Variable Name": "Begin by clearly defining what the selected
            ...   value represents|whether it's an architecture, strategy, metric,
            ...   dataset, or baseline condition. Describe exactly how this
            ...   variable will be configured, used, or operationalized in the
            ...   experiment; for example, specify how a module is implemented,
            ...   how a metric is calculated, or how a strategy is triggered.
            ...   Explain why this specific value was selected over alternatives,
            ...   including its advantages, novelty, or relevance to the
            ...   hypothesis. Describe the expected role this variable plays in
            ...   the research problem-what outcome it directly influences or
            ...   enables. If the variable is measurable, explicitly define how it
            ...   will be assessed, including the metric used, how it's
            ...   calculated, and what range of values or thresholds would
            ...   indicate a successful outcome.  If the concept is non-obvious,
            ...   include a **simple illustrative example** to aid understanding.
            ...   Your explanation should be grounded in the context of the
            ...   hypothesis and tied directly to experimental design choices and
            ...   evaluation logic.(200-400 words)",
           //Define each non-obvious technique, strategy, or mechanism used in
            ...   the hypothesis, include a 1-2 sentence example of how it would
            ...   behave in a sample input scenario.. Add detailed defintion and
            ...   description of every independent, dependent, comparable groups,
            ...   comparative variables, and control variables in simple format.
        }},

        "research_idea_design_prompt": "Describe in detail how the hypothesis
     ...   will be implemented using the agent's capabilities. If any new logic
     ...   must be built (i.e., not available as an existing codeblock),
     ...   explicitly describe how it will work at a data and control-flow
     ...   level. Explain what the new module does (e.g., filters, ranks,
     ...   reweights, scores), how it fits between existing components, and
     ...   what rules, heuristics, or computations it will use.  Describe
     ...   exactly how their outputs are linked, how data flows from one to
     ...   another, and what transformations occur at each step. + If multiple
     ...   modules or strategies are combined, explain where and how the
     ...   integration happens|in logic, in inputs/outputs, or in processing
     ...   flow. Aim for clarity so that a ASD agent could build it based on
     ...   your explanation. Include all setup steps, model configurations,
     ...   inputs/outputs expected, and how the hypothesis will be realized
     ...   end-to-end in code. (500-1000 words)",
```

```
     "research_idea_metric": "Primary and secondary metrics that will be used
...  to evaluate the hypothesis. Explain how the hypothesis will be
...  tested using concrete metrics and comparative setups. Identify the
...  benchmark tasks or datasets to be used, the control condition (e.g.,
...  a baseline agent without the component being tested), and the exact
...  performance metrics (e.g., task success rate, reasoning accuracy,
...  number of valid steps). Define how improvement or success will be
...  interpreted, including thresholds, number of runs, or statistical
...  confidence if relevant. If qualitative evaluations are involved,
...  explain how they will be derived. Ensure that all evaluations are
...  feasible using the agent's capabilities.(200-400 words)"
}},
"research_idea_name": "A short, descriptive name for the research idea (3-5
...  words)",
"research_idea_short_description": "A single concise sentence summarizing
...  the core idea (15-25 words)",
"research_baselines": "Simple list of baseline approaches to compare
...  against",
"research_idea_pilot": "Brief description of an initial small-scale test to
...  validate the approach",
"research_idea_required_code_and_resources": [
    {{
    "name": "Example Resource",
    "description": "Brief description of the resource",
    "where": "One of: 'existing codeblock', 'external', or 'build'",
    "effort": "One of: 'minor', 'moderate', or 'major'"
    }},
    // EXHAUSTIVE list of ALL required CODE, RESOURCES, MODELS, etc.
    ...  mentioned in the ENTIRE RESEARCH IDEA
        ],

"research_idea_external_requirements": [
    "example_package (for specific purpose)"
    ],

"explanation": {{
    "difference": "How it differs from the initial hypothesis",
    "novelty": "Explain exactly what is new in this configuration. Compare
    ...  it to setups or strategies found in the similar paper list. Clarify
    ...  what has not been explored and why this combination is interesting
    ...  or promising. Be specific and concise - avoid vague claims like
    ...  'this hasn't been done before'.",
    "specificity": "How is it more specific, testable, and feasible",
    "theoretical_justification": "Explain what each component does in this
    ...  experiment and why it's useful on its own. Use **concrete,
    ...  task-relevant examples**, not general claims. For instance: 'Rotary
    ...  embeddings improve recall by preserving positional clues in long
    ...  legal clauses.' Explain why any specific evaluation domain is
    ...  well-matched to the hypothesis and setup.(200-400 words)",
    "expected_synergies": "Be precise: What output from Component A is used
    ...  by Component B? Why in Condition C? At what stage? In what format?
    ...  At what decision point? E.g., 'The emotion score from module A
    ...  weights the retrieval candidates in module B before ranking.'
    ...  (200-400 words)"
}}
}}
```
```

Listing 14: Converging to a novel and testable research hypothesis given the hypothesis space

