# OpenReview forum: "HARPA: A Testability-Driven, Literature-Grounded Framework for Research Ideation"
_ICLR.cc/2026/Conference — ICLR 2026 Conference Withdrawn Submission_

### Official Review · Reviewer_XTT7 · 2025-10-29

**Soundness:** 2
**Presentation:** 1
**Contribution:** 2
**Rating:** 4
**Confidence:** 4

**Summary:**

This paper proposes a new framework named HARPA, aimed at addressing the ideation-execution gap problem that is commonly found in current large language models (LLMs) when generating scientific ideas. HARPA is mainly composed of a proposal generator and a scorer . The former simulates the workflow of human researchers to generate high-quality ideas, and the later can predict the feasibility of the proposed ideas.

**Strengths:**

The scorer mechanism is interesting. It does not rely on heuristic rules or pure LLM judgments, but learns feasibility from the actual experimental execution results. Besides, this article precisely points out a core pain point in the current Al for Science field—the disconnection between the innovativeness and feasibility of ideas.

**Weaknesses:**

1. Over-optimization for "feasibility" may suppress breakthrough innovation: The core of HARPA is to enhance the feasibility of ideas. However, a potential risk is that the system may therefore prefer those ideas that are safer, simpler, and more incremental (such as, simply replacing a network layer or testing an old model on a new dataset), while filtering out those ideas that are high-risk but may bring breakthroughs. The experimental results of the paper also partly confirm this point (HARPA's novelty score is slightly lower than the baseline). The authors should discuss this "feasibility-novelty" trade-off more deeply in the paper and explore whether the HARPA framework can be adjusted to balance or encourage higher-risk innovation.

2. The organization and readability of the paper need to be improved:
  - Frequent jumps to appendices: The narrative flow of the main text of the paper is frequently interrupted by "see Appendix X," making it difficult for readers to understand the core methods.
  - It is strongly recommended that the authors reorganize the structure of the paper, optimize the writing, and enhance the reading experience.

3. The paper lacks comparison with other methods that can be used for idea generation, such as AI-scientist [A], NovelSeek [B] and so on.
  - [A] Lu C, Lu C, Lange R T, et al. The ai scientist: Towards fully automated open-ended scientific discovery[J]. arXiv preprint arXiv:2408.06292, 2024.
  - [B] Team N S, Zhang B, Feng S, et al. NovelSeek: When Agent Becomes the Scientist--Building Closed-Loop System from Hypothesis to Verification[J]. arXiv preprint arXiv:2505.16938, 2025.

4. The scope of related work can be broader: Although the paper has cited a large number of related works, some recent highly relevant research seems to have been omitted. For example:
  - Nova: An iterative planning and search approach to enhance novelty and diversity of llm generated ideas.
  - Large language models are zero shot hypothesis proposers.
  - Large Language Models for Automated Open-domain Scientific Hypotheses Discovery.
  - Can llms generate novel research ideas? a large-scale human study with 100+ nlp researchers.
  - Dolphin: Moving Towards Closed-loop Auto-research through Thinking, Practice, and Feedback

If the author can address my concerns, I will reconsider my rating.

**Questions:**

The authors should revise the article and the figure to make the paper more readable.

---

> ### Author Response · Authors · 2025-11-21
>
> Thank you for the detailed and thoughtful review.
>
> **1. Feasibility-Novelty Trade-off.**
>
> We agree this is an important point and will expand the discussion in the revision.
>
> HARPA’s goal is to filter out logically inconsistent or non-executable ideas, not to suppress high-risk innovation. While this constraint can slightly reduce measured novelty, experts found HARPA’s proposals to remain diverse and meaningfully more actionable. Exploring alternative ways of balancing feasibility and exploratory novelty such as adjusting how strongly feasibility signals influence the generator is an interesting direction for future work, but is beyond the scope of the present study. Note, however, that the difference in novelty between the baseline and HARPA is not significant. Hence, it is unclear whether one can interpret the slight difference as less novel.
>
> Importantly, the novelty–feasibility tension is well documented. IdeaBench [A] shows that LLMs often produce highly novel but infeasible ideas, and feasibility systematically declines as novelty increases. Within this known trade-off, HARPA maintains comparable novelty while achieving substantial gains in feasibility and grounding, resulting in nearly double the ASD execution success (20 vs. 11). We will clarify that HARPA aims for **actionable novelty suitable for ASD agents**, rather than maximal unconstrained creativity.
>
> _[A] Guo, Sikun, et al. IdeaBench: Benchmarking Large Language Models for Research Idea Generation. KDD 2025._
>
> >The organization and readability of the paper need to be improved:
>
> Thank you for the suggestion. We will add the key discussions to the main text and improve the structure within the available space.
>
> >The paper lacks comparison with other methods that can be used for idea generation, such as AI-scientist [A], NovelSeek [B] and so on.
>
> Thank you for pointing out AI-Scientist [A] and NovelSeek [B]. Both [A] and [B] focus on end-to-end autonomous experimentation: they assume a candidate idea or codebase and then perform multi-round optimization, debugging, and evaluation within predefined tasks. Our work targets a different problem: open-domain, literature-grounded hypothesis-driven proposal generation. HARPA constructs structured hypotheses from arbitrary research papers, performs multi-step analysis (world-model extraction, variable–value mapping, hypothesis space exploration), and produces long-form, human-reviewable proposals. This is followed by a learned feasibility scorer trained on ASD execution traces, **capabilities not present in [A] or [B]**. Because [A] and [B] do not generate full proposals comparable to ours, and their evaluations are tied to fixed benchmark tasks, a direct quantitative comparison is not feasible. Instead, we compare it to AI-Researcher, a state-of-the-art ideation system that does produce proposal-level outputs, enabling a fair, one-to-one evaluation.
>
> We will clarify this distinction and add a related-work discussion noting that HARPA is complementary to systems like AI-Scientist and NovelSeek.
>
> Additionally, it is important to note that the feasibility settings differ. NovelSeek [B] evaluates exactly 10 ideas per fixed benchmark task (e.g., Auto2DCls, Auto3DCls) and reports executability as the number of templated experiments that successfully ran, yielding success rates such as 6/10, 7/10, or 8/10 in their tables (Table 3-4 in [B]). HARPA, by contrast, operates in an open-domain setting without predefined task templates. In our evaluation split, 45 of 120 unique proposals succeed end-to-end, reflecting the inherent variability of unconstrained idea generation. This motivates the inclusion of a learned feasibility scorer to filter low-feasibility ideas before costly execution.
>
> >The scope of related work can be broader:
>
> Please note that the **baseline used in the paper** is 'Can llms generate novel research ideas? a large-scale human study with 100+ nlp researchers.' aka AI-Researcher.
>
> We will expand the related work section to include the suggested papers. We will highlight how HARPA differs by incorporating execution-grounded feasibility prediction.
>
> >The authors should revise the article and the figure to make the paper more readable.
>
> Thank you for the suggestion to improve readability. We would appreciate guidance on which sections or figures to prioritize for clarity. This will help us make precise revisions.

---

### Official Review · Reviewer_HNiR · 2025-10-29

**Soundness:** 2
**Presentation:** 2
**Contribution:** 2
**Rating:** 4
**Confidence:** 4

**Summary:**

This paper proposes the HARPA framework, which generates testable research proposals through literature mining, hypothesis space exploration, and an executive feedback-based rater, which achieves significant improvement in feasibility and literature support.

**Strengths:**

This paper points out the problem of "disconnection between innovation and feasibility" when generating hypotheses in LLM, which is of great significance for promoting the practical application of ASD systems.

**Weaknesses:**

1. The paper proposes a complex multi-stage process, but the need for the components does not seem to be justified.
2. This paper needs to provide a detailed cost analysis and comparison.
3. Can the scorer work effectively on other ASD systems or on other scientific domains?

**Questions:**

1. How do you handle this data for the "Uncertain" class?
2. How much does reasoning trace contribute to performance?
3. How do you determine inter-rater reliability, whether different experts agree on the same proposal?

---

> ### Author Response · Authors · 2025-11-21
>
> Thank you for the detailed and thoughtful review.
>
> >How do you handle this data for the "Uncertain" class?
>
>
> Each proposal is executed multiple times in CodeScientist, and the agent produces structured summaries including a faithfulness label and a hypothesis-outcome label. As described in Section 3.2 (execution labeling), we map these into three categories: Success, Uncertain, and Failure. An execution is labeled Uncertain when the run neither cleanly succeeds nor cleanly fails, for example, when the run is partially completed, produces an ambiguous hypothesis outcome, or shows deviations without yielding a definitive error. These are cases where the agent produced some output but it is not sufficient to classify as a success, and not erroneous enough to classify as a failure.
>
> In training HARPA-Scorer, Uncertain outcomes are treated as their own category when constructing preference pairs. This ensures that ambiguous or partially successful executions do not distort the success/failure boundary, and that the scorer learns to distinguish clean feasibility signals from incomplete or inconclusive runs.
> We will clarify this in the revision.
>
>
> >How much does reasoning trace contribute to performance?
>
>
> The reasoning traces are essential. Recent work [A] has shown that feasibility assessment remains difficult: even strong LLM-based methods augmented with literature search or code generation achieve only modest improvements. In contrast, our results show that conditioning on real execution traces leads to a much larger performance gain (+0.28) in feasibility accuracy. This makes the impact of our approach clear, and we will highlight this more explicitly in the paper, especially in the context of these newer works.
>
> In our setting, the untrained LLM scorer achieves 0.52 accuracy. After distilling reasoning traces and applying preference optimization, accuracy becomes 0.81 (+0.28). The traces provide structured guidance on what factors matter for feasibility and strongly improve reliability.
>
> We also ran small exploratory tests with simpler scalar reward formulations, but they didn't give us a meaningful feasibility signal. Regressors trained on proposal embeddings collapsed to predicting the mean, and gating-style models weren't able to learn useful weights. These failures pointed to an underlying issue: feasibility is a multi-factor, reasoning-dependent signal and can't be captured by a single scalar without richer structure. This is what motivated our use of trace distillation. The distilled reasoning traces are therefore essential not only for performance but also for providing interpretable signals about feasibility. We will clarify this motivation in the revision.
>
> _[A] Jansen, Peter, Samiah Hassan, and Ruoyao Wang. "Matter-of-Fact: A Benchmark for Verifying the Feasibility of Literature-Supported Claims in Materials Science." EMNLP (2025)._
>
>
> >How do you determine inter-rater reliability, whether different experts agree on the same proposal?
>
>
> In our setup (as explained in Section 4.2), each proposal set was reviewed by the same expert who selected the source paper from their own research area. As a result, proposals were unique to each expert and not cross-reviewed. This keeps the HARPA vs. baseline comparison within a single, domain-appropriate reviewer, ensuring evaluations remain consistent within the relevant expertise. Prior work [A] shows that reviewers evaluating ideas outside their own field often produce inconsistent scores so keeping each proposal set with a domain-appropriate expert is a reliable and well-supported choice.
>
> _[A] Boudreau, Kevin J. et al. “Looking Across and Looking Beyond the Knowledge Frontier: Intellectual Distance, Novelty, and Resource Allocation in Science.” Management science 62 (2016): 2765 - 2783._
>
>
> >Can the scorer work effectively on other ASD systems or on other scientific domains?
>
>
> Yes. The scorer is conditioned on an explicit agent profile (capabilities, constraints). This design is agent-agnostic: adapting it to a new ASD system only requires execution traces from that system. The mechanism generalizes; only the data source changes.

---

### Official Review · Reviewer_1ifE · 2025-10-31

**Soundness:** 2
**Presentation:** 3
**Contribution:** 3
**Rating:** 4
**Confidence:** 4

**Summary:**

This paper proposes a scientific research hypothesis generation framework named HARPA, aiming to address two major challenges of large language models (LLMs) in scientific research creativity generation: ensuring the testability and literature-groundedness of hypotheses. The HARPA framework consists of two core components: (1) a multi-stage hypothesis generator (Proposal Generator), which identifies research trends, constructs hypothesis space through literature mining, and finally converges to specific hypotheses that fill research gaps; (2) a hypothesis scorer (Proposal Scorer), which is a reward model (RM) trained based on previous experimental execution trajectories (success or failure) and used to predict the feasibility of new hypotheses.

The authors evaluated HARPA through two sets of experiments: (1) Human expert evaluation (compared with the AI-Researcher baseline), which showed that HARPA was significantly superior in "feasibility" and "groundedness"; (2) Automated scientific discovery (ASD) agent evaluation, which showed that the hypotheses generated by HARPA had a higher execution success rate (20 vs 11) on the ASD agent (CodeScientist). Additionally, an independent experiment demonstrated that HARPA's scorer was 28% more accurate than the untrained LLM baseline in predicting execution success rates.

**Strengths:**

1. Important Issue: This paper addresses a critical and timely issue: how to bridge the gap between the "creative generation" and "actual scientific research execution" of LLMs.
2. Execution-oriented rewards: Taking "feasibility" as a key indicator and attempting to use actual ASD agent execution trajectories (not just zero-shot judgments from LLMs) to train the reward model is the right and valuable direction.
3. Breadth of Evaluation: The evaluation design of the paper (although flawed) attempts to cover both human expert evaluation and real agent execution simultaneously, and this multi-dimensional evaluation approach is commendable.

**Weaknesses:**

1. Lack of end-to-end evaluation: There is a lack of critical end-to-end (End-to-End) verification. The paper claims that its core advantages lie in "testability-driven" and "Self-Adaptation to prior experimental outcomes". This strongly implies a closed-loop system: the feedback from the Scorer should be able to guide or improve the Generator. However, the experimental design in this paper is completely disjointed:
    * Experiment 5.1 evaluated the generator (compared to AI-Researcher).
    * Experiment 5.2 evaluated the scorer (compared to the untrained LLM).
    * Missing Experiment: The authors never conducted an end-to-end evaluation to prove that the combination of "generator + scorer" outperforms the "generator (alone)". The authors only demonstrated that they could train a scorer, but never used this scorer in experiments to filter or re-rank the generator's output and verify whether doing so (e.g., taking the Top-K) could further improve the ratings of human experts or the execution success rate of ASD agents. This leaves the paper's core claims of "Self-Adaptation" and "learning from experience" without the most direct experimental verification.
2. Unfair baseline comparison: There are serious confounding variables in the core human evaluation and agent evaluation in Section 5.1. The input of HARPA (full paper) is far more informative than the base line (topics generated from abstracts). This makes it impossible for us to attribute the observed improvements in "feasibility" and "groundedness" to the HARPA framework, and it may simply be due to the difference in input information. This fundamentally weakens the argument for the effectiveness of the HARPA generator. This is manifested in at least two aspects:
    * (a) Input Inconsistency: As described in Section 4.2, the input of HARPA is the "source paper", while the input of the base line AI-Researcher is the "topic generated from the abstract of each source paper". A complete paper clearly provides far richer and more specific context than a single topic word. The significant advantages of HARPA in "groundedness" (+0.85) and "feasibility" (+0.78) are most likely solely due to this difference in input granularity, rather than the superiority of its generator process itself.
    * (b) Uncontrolled Retriever: The paper acknowledges that HARPA and AI-Researcher each used their own internal literature retrieval processes. Literature retrieval is the lifeblood of "groundedness". The authors did not control this variable (e.g., having both systems use exactly the same retrieval results as input), making it impossible for us to determine whether the performance improvement comes from HARPA's novel generation process or simply from a (possibly superior) literature retrieval component.
3. Sacrificed novelty: This is a key issue. According to Table 6 (Appendix), in the human evaluation, HARPA's "Novelty" score (5.98) is actually lower than the base line (6.43). Although the authors stated in the main text that "novelty is not sacrificed", for a top-tier conference like ICLR, a paper that (possibly defectively) excels in feasibility but is on par or even lags behind in novelty has limited contributions.
4. Generalization Ability and Domain Mismatch: The Scorer of HARPA was trained on a dataset of ACL (NLP domain) papers (Section 4.3). However, the evaluation of the papers (Section 4.2) covers a broader range of topics, including "RL (reinforcement learning), Optimization". The authors did not demonstrate whether this "feasibility" predictor trained on NLP can generalize to distinct domains such as RL or optimization. The universality of this framework for scientific domains other than those tested in the paper has not been fully explored.
5. Failure to address complex issues: The methodology and evaluation of the paper seem to focus on relatively straightforward hypotheses that can be reduced to "key variables". The paper does not explore how HARPA addresses multi-faceted research questions that require multi-step experiments or involve complex interactions among multiple variables.
6. Analysis of Lack of Computational Resources: The paper does not conduct a detailed analysis of the computational resources required for training and executing the HARPA framework (including the generator and scorer). This makes it difficult for other researchers to evaluate the feasibility of reproducing or deploying the framework in their laboratories.

**Questions:**

1. (Regarding Weakness 1): Why didn't the author conduct end-to-end evaluation? For example, using the trained HARPA-Scorer to re-rank the proposals generated by HARPA-Generator, and then submitting the top-K proposals with the highest scores to human experts and ASD agents. This seems to be a direct way to verify the actual utility of the scorer and is also crucial to support your "test-driven" claim.
2. (Regarding Weakness 2): How can the author prove that the improvements in feasibility (+0.78) and groundedness (+0.85) observed in Section 5.1 stem from HARPA's superior generation process, rather than simply because HARPA was given a much more informative input (full paper vs. abstract topic)?
3. (Regarding Weakness 3): The evaluation in Table 6 shows that HARPA is lower than the base line in terms of novelty. Does this mean that the framework sacrifices novelty in exchange for (possibly problematic) improvements in feasibility? This seems to be a significant compromise for a "creative generation" tool.
4. (Regarding Weakness 4): The HARPA scorer is trained on NLP (ACL) data. How accurate is it when evaluating the feasibility of proposals in non-NLP domains (such as RL or optimization)? Are there out-of-distribution generalization (OOD) issues?
5. (Regarding Weakness 5): The method of the paper focuses on extracting variables and values (A + B). How does it handle more complex research questions that require multi-step experiments or involve complex interactions (non-direct causality) between variables?
6. (Regarding Weakness 6): Could the author provide a detailed analysis of the computational resources required for the training and inference of HARPA (including the generator and scorer)?

---

> ### Author Response · Authors · 2025-11-21
> **Official Comment by Authors**
>
> Thank you for the detailed and thoughtful review.
>
> (Our response is in multiple parts because of length limits.)
>
>
> **1. Regarding Weakness 1: end-to-end evaluation**
>
> Thank you for highlighting the value of a full end-to-end "Generator --> Scorer --> Top-K proposals" evaluation.
>
> In our case, we didn't run an end-to-end study mainly because of resource limits. As we mentioned in Section 4.3, running such a pipeline would require generating many additional proposals per source paper and then executing all of them with ASD agents, plus collecting expert annotations. Prior work such as  CodeScientist [A] shows that ASD-based execution is resource-intensive even for individual runs.
>
> There is also a sequencing issue: the scorer is only meaningful once the generator can reliably produce specific, executable hypotheses; otherwise, there is no reliable feasibility signal to learn from. Because of that dependency, we focused first on evaluating the two components separately. We therefore run an expert study for the ideation component (with enough samples to clearly see the benefits of using the proposed generator versus the baseline method), and an ASD-based feasibility evaluation for the scorer. This setup isolates their contributions and verifies that (i) the generator produces grounded, testable hypotheses, and (ii) the scorer can accurately model execution feasibility from  execution traces from HARPA proposals.
>
> We agree that an end-to-end study is the natural next step, and we will make that clearer in the revision.
>
> _[A] Jansen, Peter, et al. "Codescientist: End-to-end semi-automated scientific discovery with code-based experimentation." Findings of the Association for Computational Linguistics: ACL 2025. 2025._
>
>
> **2a. Regarding Weakness 2(a) ("input mismatch concern").**
>
> Both systems start from the same underlying textual information, the abstract of the source paper. HARPA does not ingest the full source-paper. When we say "starting from a source paper," we mean using its title and abstract as seed context, exactly as shown in Appendix L (Listing 8, L2757). The LLM never sees the full paper; instead, HARPA relies on its own retrieval logic to gather additional literature.
>
> AI-Researcher, by design, expects a topic string as input, which is why we follow its original usage and provide a topic derived from the same abstract (Section 4.1). However, it takes a topic string only as a seed. It then runs a broad literature search over the full Semantic Scholar index, gathers many candidate papers, and ranks them using its own retrieval logic.
>
> Since both pipelines are seeded with equivalent input content, the observed gains in feasibility and grounding come from HARPA's multi-stage reasoning process, not from receiving more information.
>
> We will clarify this to avoid misinterpretation.
>
>
> **2b. Regarding Weakness 2(b) (uncontrolled retriever).**
>
> We agree that both AI-Researcher and HARPA have different retrieval procedures - both of which operate over the same underlying information: the Semantic Scholar index. However, it is exactly the difference in retrieval, which in both cases is an integrated and core part of the algorithms of the two systems. Since we are comparing the two systems end-to-end on the same literature universe and under the same expert-evaluation protocol, it is appropriate that each uses its own retrieval approach.
>
> To clarify the input setup:  HARPA is not given the full source paper as input; both systems start from abstract-derived context and gather additional literature through their own retrieval pipelines. AI-Researcher performs iterative LLM-driven retrieval _(KeywordQuery, PaperQuery, GetReferences)_, while HARPA uses structured citation-chain expansion and snippet search. We will make this point more explicit in the revision.

---

> ### Author Response · Authors · 2025-11-21
>
> (continued from previous comment)
>
> **3. Regarding Weakness 3 (novelty).**
>
> HARPA's novelty score $(5.98 \pm 1.33)$ is modestly below the baseline's $(6.43 \pm 1.32)$, but this difference is **not statistically significant** (bootstrap p = 0.937; Wilcoxon p = 0.107; Table 7). According to the evaluation rubric used in the study (Appendix B, L715), a score of **5 corresponds to "somewhat novel / incremental" and 6 corresponds to "reasonably novel"**. Both systems fall within this same range of incremental but meaningful novelty. Hence, (statistically) we cannot say that HARPA's output is less novel than the one of baselines and our evidence does not show that HARPA enters a different tradeoff.
>
> Purely from a qualitative point of view, the baseline ideas sometimes appeared more "novel" because they were less specific or more open-ended, whereas HARPA's proposals were more operational and grounded in retrieved evidence, which naturally limits purely speculative creativity. Crucially, HARPA maintains comparable novelty while delivering significant improvements in **feasibility (+0.78, p < 0.05) and grounding (+0.85, p < 0.01)**, and this corresponds to substantially higher ASD execution success (20 vs. 11).
>
> Generally, beyond our comparison, we would like to note that the novelty-feasibility tension is well documented in prior work. For example, IdeaBench [A] shows that LLMs often produce highly novel but infeasible ideas, and feasibility consistently drops as novelty increases. In that context, HARPA’s ability to maintain comparable novelty and improve feasibility and grounding is exactly what makes it useful for ASD agents. We will clarify in the revision that HARPA targets actionable novelty suitable for ASD agents, not maximal speculative novelty.
>
> _[A] Guo, Sikun, et al. "Ideabench: Benchmarking large language models for research idea generation." Proceedings of the 31st ACM SIGKDD Conference on Knowledge Discovery and Data Mining V. 2. 2025._
>
>
> **4. Regarding Weakness 4 (OOD generalization of the scorer).**
>
> The HARPA-Scorer is trained on execution traces obtained from running proposals derived from ACL papers (source paper abstracts) on the ASD agent CodeScientist (Section 3.2). These traces contain structured summaries, error logs, and success/failure signals, so the scorer mainly learns patterns about what the ASD agent can or cannot execute (e.g., supported libraries, resource limits, restricted operations).
>
> Because the execution-trace supervision comes from proposals derived from ACL papers (abstracts), our empirical evaluation of the scorer is restricted to this distribution. We do not claim validated accuracy in non-NLP domains (such as RL or optimization). **As a small robustness check, we also evaluated 35 proposal pairs from the human-evaluation corpus, which includes a broader mix of computational topics (Appendix Table 4). The scorer achieved 74% accuracy under the same preference-pair criterion;** while this does not establish cross-domain generalization, it may suggest that the model is not exclusively tuned to ACL-only topics. We will add these additional results and explicitly state this limitation in the revision.
>
> However, the design itself is general: the scorer conditions on the ASD agent's explicit profile and could be adapted to other domains or other ASD agents if execution traces from those settings become available. Demonstrating such cross-domain extensions would require collecting domain-specific execution traces, which we see as natural future work. We will clarify this intended scope in the revision.
>
> **5. Regarding Weakness 5 (handling multi-step or complex research questions).**
>
> HARPA's variable-value extraction step structures the hypothesis space but does not constrain the generator to single-variable or single-step substitutions. The variable-value graph is provided as grounded context only, and the LLM remains free to synthesize multi-component or multi-step hypotheses; the generation prompts **(Appendix F, e.g., Line 3268)** explicitly encourage multi-step reasoning before proposal generation.
>
> For example, in Appendix C, the final proposal contains multiple interacting mechanisms, multi-stage methodological designs, and multi-part evaluation pipelines. These demonstrate that HARPA is not limited to direct variable-level swaps; the extraction stage serves only to ensure each component is literature-supported and testable.
> We will clarify this in the revision.

---

> ### Author Response · Authors · 2025-11-21
>
> (continued from previous comment)
>
> **6. Regarding Weakness 6 (computational resources).**
>
> We already provide the main configuration details for HARPA in the paper: Appendix D.1 lists the scorer's model size, hardware setup, sequence length, batch configuration, and so on. The scorer is trained in two stages: (i) a single-epoch distillation run on 4×A100-80GB GPUs, and (ii) a single-epoch RLVR fine-tuning run on 8×A100-80GB GPUs. What was missing from the appendix is the run-time requirements (our apologies): (i) required 3h 8m 28s and (ii) 8d 10h 11m 47s of compute. HARPA-Generator does not require model training; as noted in Line 217, both HARPA and the baseline systems use fixed multi-stage LLM pipelines invoked via standard API inference, so no additional training compute is involved.
>
> Scorer training is a one-time cost, but constructing its training dataset is expensive because each supervision signal requires a full ASD-agent execution (code synthesis + experiment run). Section 4.3 describes this execution process and the scale of runs needed to construct the dataset. Once trained, scorer inference is lightweight: it runs locally on a 7B model and is negligible compared to executing a full CodeScientist experiment.
>
> The benefit becomes clear when looking at the evaluation split:  it contains **120 unique proposals**, of which only **45** succeed and **75** fail when executed end-to-end. With **81%** accuracy, the scorer can correctly flag roughly **61 of these 75** low-feasibility proposals before execution. Because a single ASD-agent execution is orders of magnitude more expensive than scorer inference, skipping most of these failed runs drastically reduces end-to-end cost when scaling to many ideas.
>
> We will make these pointers more visible in the revision and add the run times.

---

### Author Response · Authors · 2025-12-03

We thank all reviewers for their helpful and constructive comments.

We appreciate the reviewers' comments recognizing the importance of the problem and the value of grounding feasibility in real ASD execution trajectories for practical ASD systems. We are encouraged by the reviewers' acknowledgement of the breadth of our evaluation, which combines human expert assessment with actual experimental execution results.

Below, we responded to individual points in detail, and we have also uploaded a revised PDF with the following high-level edits and clarifications:

* **Clarified the input setup.** HARPA does not ingest full papers. We now state clearly in the main text that only the abstract was used as input, addressing the misunderstanding behind several concerns.
* **Expanded discussion of the novelty-feasibility trade-off.** We highlighted that the novelty difference is not statistically significant and clarified HARPA's goal of producing actionable novelty. We also add context from prior work on the expected trade-off between novelty and feasibility.
* **Addressed all minor clarifications** including computational details, execution-labeling (Uncertain class), reasoning-trace contribution, and OOD limitations.
* **Expanded related work.** We included the recent systems highlighted by reviewers (e.g., NovelSeek, Nova, Dolphin) and clarified how they differ from HARPA's focus.

---

### Note · Authors · 2026-01-04

**Comment:**

We would like to withdraw this submission to submit a revised version to another venue.

**Withdrawal Confirmation:**

I have read and agree with the venue's withdrawal policy on behalf of myself and my co-authors.